



# Nitrification and Nitrite Isotope Fractionation as a Case Study in a major European River

Juliane Jacob[1, 2][*], Tina Sanders[1], Kirstin Dähnke[1]

[1] Helmholtz Center Geesthacht, Institute for Coastal Research, Geesthacht, Germany
[2] University of Hamburg, Institute of Biogeochemistry and Marine Chemistry, Hamburg, Germany

*Correspondence to*: Juliane Jacob (juliane.jacob@hzg.de)

**Abstract.** In oceans, estuaries, and rivers, nitrification is an important nitrate source, and stable isotopes of nitrate are often used to investigate recycling processes in the water column. The bulk isotope effect of nitrification is hard to predict: It is a two-step-process, where ammonia is oxidized via nitrite to nitrate. Nitrite usually does not accumulate in natural environments, which makes it even more difficult to unravel the divergent isotope effects of both processes.

However, during an exceptional flood in the Elbe River in June 2013, ammonium and nitrite accumulated in the water column for a short period, returning towards normal summer conditions within one week. Concentrations were sufficient for the analysis of $\delta^{15}N-NH_4^+$ and $\delta^{15}N-NO_2^-$ evolution, which has not been studied before in a major European river like the Elbe River. In the concert with changes in SPM and $\delta^{15}N$-SPM, as well as nitrate concentration, $\delta^{15}N-NO_3^-$ and $\delta^{18}O-NO_3^-$, we calculated the isotope fractionation effect during nitrification. We found that in the water column, ammonium and nitrite derived from internal recycling processes, whereas nitrate mainly leached from catchment area. Ammonium and nitrite concentrations increased to 3.4 µmol L$^{-1}$ and 4.5 µmol L$^{-1}$, respectively, due to remineralization and ammonium oxidation in the water column. $\delta^{15}N-NH_4^+$ values increased up to 12‰, and $\delta^{15}N-NO_2^-$ ranged from -8.0‰ to -14.2‰. As water column nitrite concentration decreased, we calculated an isotope effect $^{15}\varepsilon$ of -9.3‰ for nitrite oxidation. This isotope effect does not correspond to the inverse isotope fractionation with a positive $^{15}\varepsilon$ proposed by pure culture studies. We hypothesize that the molecular mechanisms that lead to inverse fractionation also apply in natural environments, but that the resulting trend in $\delta^{15}N-NO_2^-$ in this natural environment is masked by dilution with fresh nitrite stemming from ammonium oxidation.

Our data are a first approximation of the isotope effect of nitrite oxidation in natural environments and highlight that pure culture results cannot readily be extrapolated to natural microbial assemblages or water bodies.

## 1 Introduction

Today's nutrient input to aquatic systems is significantly elevated over pristine background values in rivers and estuaries all over Europe. Since 1860, the input of reactive nitrogen ($N_r$) increased 20-fold to about 150 Tg N yr$^{-1}$ (Galloway and Cowling, 2002). The resulting eutrophication and its impacts have been discussed extensively (e.g. (Rabalais, 2002; Galloway et al., 2003; Smith et al., 2006). In 1985, North Sea bordering countries decided to reduce nutrient inputs by 50%. As a result, the overall water quality, and especially DIN (dissolved inorganic nitrogen) loads as well as oxygen saturation have improved markedly (Pätsch et al., 2010). Ammonium inputs to the Elbe



River decreased by 93% from 1986 to 2006 (Bergemann and Gaumert, 2008), because of an improved waste water and organic carbon management. Today, the riverine DIN load consists mainly of nitrate stemming from urban waste water, surface runoff, and leachate from agriculture soils (van Breemen et al., 2002; Brion et al., 2004).

Nevertheless, nitrate regeneration in rivers can also modify DIN loads (Middelburg and Nieuwenhuize, 2001):
Remineralization of organic material and subsequent nitrification (Mayer et al., 2001) regenerates nitrate, which then again enters the nitrogen cascade (Galloway et al., 2003) and can either be denitrified (Mariotti et al., 1981; Böttcher et al., 1990; de Wilde and de Bie, 2000) or assimilated by bacteria and phytoplankton (Wada and Hattori, 1978; Middelburg and Nieuwenhuize, 2000). Nitrate regeneration via nitrification occurs in major rivers throughout Europe, and contributes to nitrate loads in, for example, the Seine, Scheldt and Elbe Rivers (de Wilde and de Bie,
2000; Sebilo et al., 2006; Johannsen et al., 2008). A previous study by Johannsen et al. 2008 suggested that in the contemporary Elbe River, nitrate derived from nitrification in soils was the main constituent of the water column nitrate load in winter.

During biological nitrogen uptake, the isotope composition of the source nitrogen changes because biological processes usually favour the light isotope over the heavy ones (Mariotti et al., 1981; Kendall, 1998). Based on a
closed-system Rayleigh distillation model, the fractionation factor $^{15}\varepsilon$ for nitrogen uptake can be calculated; individual uptake processes have specific fractionation factors (Rayleigh, 1896; Broecker and Oversby, 1971).

During nitrification, a further obstacle is that it is a two-step-reaction with divergent isotope effects. Wide ranging fractionation factors of -14 to -34‰ occur during the first step, ammonia oxidation to nitrite, in different pure cultures (Delwiche and Steyn, 1970; Mariotti et al., 1981; Casciotti et al., 2003). The second step, the oxidation of
nitrite to nitrate, exhibits very rare inverse fractionation (Casciotti, 2009; Buchwald and Casciotti, 2010): The newly produced nitrate is heavier than the source nitrite, and remaining nitrite in turn gets subsequently depleted during nitrite oxidation. After complete consumption, the isotope value of the substrate is equal to the product.

This illustrates that the interpretation of isotope changes in natural environments during nitrification is complex, and studies addressing the combined fractionation factor of ammonium and nitrite oxidation together even in culture are
scarce. Moreover, investigations of nitrite oxidation and its isotope effect in natural environments are hampered by the fact that nitrite concentrations in relatively healthy and actively nitrifying environments are too low to analyze isotope values.

This is also the case in the Elbe River: Under normal flow conditions, nitrite is not abundant; the main DIN species is nitrate, which shows a distinct seasonal cycle. Nitrate concentration in winter rises above 300 µmol L$^{-1}$; summer
values are in the range of ~80 µmol L$^{-1}$ due to intense biological nitrate uptake (Johannsen et al., 2008; Schlarbaum et al., 2011). The interplay of isotopically distinct nitrogen sources and fractionation processes also leads to distinct summer and winter nitrate isotope values in the water column. Isotope values are highest in summer due to biological uptake and phytoplankton production (Van Beusekom and De Jonge, 1998), and lowest in winter (Johannsen et al., 2008; Schlarbaum et al., 2011). The annual mean d$^{15}$N-NO$_3^-$ value is 8.5‰, which is typical for catchment areas with
more than 60% of agricultural and urban land use (Grischek et al., 1998; Johannsen et al., 2008).

These normal hydrological conditions were disrupted by an unusual summer flood in the Elbe River in June 2013 (Fig. 1). Hydrological conditions changed and runoff and turbidity increased drastically. Phytoplankton is light dependent and should thus be adversely affected by turbidity, but nitrifiers are not. We expected high turbidity and



temperature to provide optimum conditions for nitrifiers, so that ammonium and nitrite concentration changes can be attributed to nitrification. Indeed, concentration of nutrients, especially nitrite, rose quickly, which was a unique opportunity to analyze isotope changes. To the best of our knowledge, this is the first investigation of isotope fractionation during nitrite oxidation in a natural, actively nitrifying, river system.

## 2 Materials and Methods

### 2.1 Study site

Nearly 25 million people live in the catchment area of about 148,000 km$^2$ of the Elbe River. After the Rhine River, the Elbe is the second largest stream discharging into the North Sea and the largest source of nitrate and DIN for the inner German Bight (Brockmann and Pfeiffer, 1990). The average discharge is about 738 m$^3$ s$^{-1}$ with an annual discharge of 23 km$^3$ (Lozán et al., 1996) and a nitrate load of about 76 kt yr$^{-1}$ (Bergemann and Gaumert, 2010). Ammonium is of minor importance and is <5% of the nitrate load, whereas nitrite is usually <2%.

Our study site at stream kilometre 585 is located upstream of a weir that separates the river from the tidal estuary (53°25′31′′N, 10°20′10′′E). Discharge was measured upstream at the nearest gauge at Neu Darchau, stream kilometre 536.5.

### 2.2 Sampling

During the flood event in June 2013, surface water samples were taken twice a day from 6 to 14 June and, with decreasing discharge, daily on 15, 16, 18, and 20 June. Water temperature was measured immediately after sampling and samples were transferred into 2L PE bottles for immediate processing. Water samples were filtered within an hour (preweighed GF/F, precombusted at 450°C, 4.5 hrs), and aliquots of filtered water samples were frozen for later nutrient concentration analyzes, and stable isotope composition ($\delta^{15}$N-NH$_4^+$, $\delta^{15}$N-NO$_2^-$, $\delta^{15}$N-NO$_3^-$, $\delta^{18}$O-NO$_3^-$). Filter samples were dried at 50°C and weighed for later determination of SPM content, and $\delta^{15}$N-SPM analysis.

Nutrient concentrations were analyzed with a continuous flow analyzer (AA3, Seal Analytics, Germany). For nitrite and nitrate standard photometric techniques were used (Grasshoff et al., 2009), and ammonium was measured fluorometrically with a detection limit of 0.5 µM based on Holmes et al. (1999).

### 2.3 Isotope analyzes

Dual nitrate and nitrite isotopes were analyzed using the denitrifier method (Sigman et al., 2001; Casciotti et al., 2002). In brief, water samples were injected into a concentrated *Pseudomonas aureofaciens* (ATCC#13985) suspension to analyze nitrate and nitrite. Nitrite concentration was always <2% of nitrate in water samples. For separate analysis of the isotopic signature of nitrite, *Stenotrophomonas nitrireducens* bacteria were used to selectively reduce nitrite (Böhlke et al., 2007). Both bacteria denitrify the substrate to N$_2$O gas, which is then analyzed on a GasBench II, coupled to a Delta V isotope ratio mass spectrometer (Thermo Fisher Scientific). The sample volume was always adjusted to achieve identical gas amount in the samples (final gas amount of 10 nmol in case of nitrate, 5 nmol for nitrite analysis) to avoid concentration-dependent fractionation effects.





For analysis of the ammonium isotopic composition, ammonium was chemically converted to nitrite and then reduced to $N_2O$ using sodium azide (Zhang et al., 2007). Ammonium isotopes were analyzed in all samples with $[NH_4^+] > 1$ µmol L-1. Sample gas extraction and purification was equivalent to nitrite and nitrate isotope samples.

$\delta^{15}N$-SPM of suspended matter was analyzed with an element analyzer Carlo Erba NA 2500 coupled with an isotope ratio mass spectrometer Finnigan MAT 252.

Isotope values are reported using the common "delta" notation (cf. Eq. 1) (McKinney et al., 1950),

$$\delta^{15}N \ [‰ \ vs. \ std] = \left( \frac{\left( \frac{^{15}N}{^{14}N} \right)_{sample}}{\left( \frac{^{15}N}{^{14}N} \right)_{std}} - 1 \right) * 1000 \qquad (1),$$

where the standards for nitrogen and oxygen are atmospheric $N_2$ and Vienna Standard Mean Ocean Water (VSMOW), respectively, which both by definition have a $\delta$-value of 0‰.

International solid secondary standards with known $\delta^{15}N$-values were used for calibration. IAEA N3, USGS 34 and an internal potassium nitrate standard were used for nitrate isotope analysis; IAEA N1, IAEA N2, and a certified sediment standard (IVA Analyzentechnik, Germany) for suspended matter isotope values; and IAEA N1, USGS 25, and USGS 26 were used to calibrate ammonium isotope values. For nitrite isotope analysis, we used in-house potassium nitrite and sodium nitrite standards with known $\delta^{15}N$ values of -81.55‰ and -27.46‰, determined via EA/IRMS analysis. All samples were analyzed in duplicate to calculate standard deviations. Standard deviation of reference material was <0.2‰ for $\delta^{15}N$-$NO_3^-$ and <0.5‰ for $\delta^{18}O$-$NO_3^-$. For nitrite isotope analysis, the standard deviation of $\delta^{15}N$-$NO_2^-$ was <0.3‰, and that of $\delta^{15}N$-$NH_4^+$ was <0.5‰. The standard deviation of $\delta^{15}N$-SPM was <0.1‰. For quality assurance, additional internal standards were analyzed in every run.

The fractionation factor $\varepsilon$ can be calculated based on the Rayleigh distillation equation of a closed-system model (Broecker and Oversby, 1971; Mariotti et al., 1981) as

$$\varepsilon_{p/s} = \frac{\delta_s - \delta_{s,0}}{\ln f} \qquad (2)$$

Where p is the product, s is the substrate, $\delta_s$ and $\delta_{s,o}$ is the delta value of substrate at the time of sampling and the initial value, respectively. $f$ is the remaining fraction of substrate at the time of sampling.

### 3 Results

#### 3.1 General hydrographic properties

Flood conditions with discharge values >3000 $m^3$ $s^{-1}$ at gauge Neu Darchau (J. Kappenberg, pers. comm.) occurred from 9 to 18 June due to extremely high precipitation and resulting runoff in the catchment area. On 11 and 12 June, maximum SPM values of 70 mg $L^{-1}$ were eluted shortly before peak discharge with 4060 $m^3$ $s^{-1}$ and decreased afterwards to 9.2 mg $L^{-1}$ (Fig. 2a). Throughout the entire flood, the water temperature was high and increased from 16.2 to 21.5°C.

Dissolved oxygen concentration was clearly correlated to discharge; the concentration was initially about 10 mg $L^{-1}$, corresponding to an oxygen saturation of about 100% and more, depending on the time of day of sampling. With increasing discharge, the oxygen concentration droped to a minimum of 6.0 mg $L^{-1}$ (corresponding to 63%




saturation), before increasing again to an intermediate maximum of 7.6 mg L$^{-1}$. After this peak, [$O_2$] decreased, accompanied by a strong increase in water temperature (Fig. 2a).

### 3.2 Nutrient concentrations

Previous studies (Johannsen et al., 2008; Schlarbaum et al., 2011) found higher nutrient concentrations in winter and
lower in summer seasons. Our data generally are in line with their findings, but appear more representative of spring than of summer conditions, because winter and spring 2015 were unusually cold (Van Oldenborgh et al., 2015), so that a slight seasonal offset must be taken into account. Discharge was >800 m$^3$ s$^{-1}$ and nitrate concentrations was >200 µmol L$^{-1}$ before the flood. Nitrite concentration was <1.2 µmol L$^{-1}$, and ammonium concentration was below the detection limit of 0.5 µM.
The DIN concentration increased when discharge rose to >3000 m$^3$ s$^{-1}$ and reached a distinct maximum shortly after peak discharge (Fig. 2b). At high discharge (>3000 m$^3$ s$^{-1}$) nitrite concentration rose above >2.2 µmol L$^{-1}$ and, along with all other nutrients, reached a maximum of 4.4 µmol L$^{-1}$ on 14 June, followed by a decrease to 3.3 µmol L$^{-1}$ towards the end of the flood event (Fig. 2b).

Ammonium concentration rose above the detection limit and reached a maximum of 3.2 µmol L$^{-1}$ immediately after
the peak of SPM and oxygen concentrations below 7 mg L$^{-1}$ (Fig. 2b). The elevation of nitrite concentration above 2.7 µmol L$^{-1}$ is coupled to a decrease in oxygen (<7.6 mg L$^{-1}$).

With decreasing discharge, the oxygen concentration rose, ammonium concentration dropped below the detection limit, and the overall DIN concentration decreased again (Fig. 2).

### 3.3 Isotope trends of DIN and of particulate nitrogen

During the entire flood (excluding discharge below 2000 m$^3$ s$^{-1}$), $\delta^{15}$N-NO$_3^-$- and $\delta^{18}$O-NO$_3^-$-values are negatively correlated with discharge and nitrate concentration (R$^2$ = 0.8 and 0.5, respectively, not shown in plots). The range of δ-values of nitrate during the flood is relatively narrow: Initial values of $\delta^{15}$N-NO$_3^-$ and $\delta^{18}$O-NO$_3^-$ are 9‰ and 3.5‰, respectively, dropping to 7.4 and 2.1‰ when nitrate concentration is highest (Fig. 2c). Afterwards, δ-values of nitrate increase again, alongside with dropping concentration, reaching values of 8.8 and 3.9‰ for $\delta^{15}$N-NO$_3^-$ and
$\delta^{18}$O-NO$_3^-$, respectively.

The nitrite isotope values follow a complex pattern when compared to nitrite concentrations (Fig. 2b, Fig. 2c). Before the flood, nitrite concentration increased slightly, while $\delta^{15}$N-NO$_2^-$ increased from -14.2 to -8.0‰. At higher discharge (>2000 m$^3$ s$^{-1}$), nitrite concentration quickly rose to a maximum of 4.4 µmol L$^{-1}$, while $\delta^{15}$N-NO$_2^-$ decreased from -8.0 to -13.8‰. When discharge decreased, nitrite concentration also decreased, coupled to a clear
increase of $\delta^{15}$N-NO$_2^-$, which corresponds to a calculated fractionation factor ε of -9.3‰ with R² 0.98377 (Fig. 4, Eq. 2).

At the beginning of the flood event, ammonium concentration rose, so that $\delta^{15}$N-NH$_4^+$ could be analyzed. Shortly after the SPM peak, $\delta^{15}$N-NH$_4^+$ is about 2‰ and then increases with time to a maximum of 12‰ shortly after peak discharge, followed by a decrease to about 6‰. Although the lowest isotope value coincides with minimal
ammonium concentration, there is no distinct correlation of ammonium concentration and its isotope composition. Overall, $\delta^{15}$N-NH$_4^+$ seemed to be weakly correlated to the SPM load, but not so much to $\delta^{15}$N-SPM: The changes in




$\delta^{15}$N-SPM, though ranging from 8.1 to 6.2‰ during the flood event, were minimal at the time of ammonium occurrence. It is interesting, however, that initial $\delta^{15}$N-$NH_4^+$ values were about 4‰ lighter than suspended matter.

## 4 Discussion

### 4.1 Nutrient dynamics and evidence in isotope changes during the flood

The high terrestrial runoff from the catchment area results in an SPM peak from terrestrial sources, which is eluted directly before the discharge peak. This succession is typical for flood events and has been observed in the Elbe River before (Baborowski et al., 2004; Pepelnik et al., 2005).

Nitrate concentration initially decreased with increasing discharge, because the river nitrate load was diluted with high amounts of precipitation and terrestrial runoff. After 10 June, nitrate concentration increased with discharge, along with the SPM peak in the river. Both can be attributed to leachate from agricultural soils: terrestrial soil nitrate, leached due to high precipitation in the catchment area, is an important nitrate source to the river system at this time of the year (Johannsen et al., 2008), and during the flood, leached SPM and nitrate are extraordinarily high and clearly shown in the water mass. The decrease of $\delta^{15}$N-SPM values from 7.8 to 6.2‰ during highest discharge also indicates the input of fresh organic material due to leachate of unfractionated material from the catchment area.

The effect of biological processing and assimilation on the nitrate pool can be inferred from concentration and isotope changes after the SPM peak and maximal concomitant nitrate input. Under normal conditions, biological activity is low in winter, nitrate concentration is high and isotope values are low, because no uptake and hence no fractionation occur. In the Elbe River typical winter $\delta^{15}$N-$NO_3^-$ and $\delta^{18}$O-$NO_3^-$ values are 7.8 – 9.3 and 0.8 ‰, respectively (Johannsen et al., 2008; Schlarbaum et al., 2011). During the flood in June, we see similar values: $\delta^{15}$N-$NO_3^-$ ranges between 7.4 – 9‰ and $\delta^{18}$O-$NO_3^-$ between 2.1 – 3.9‰ (Fig. 2c). The narrow ranges and low values of $\delta^{15}$N-$NO_3^-$ and $\delta^{18}$O-$NO_3^-$ indicate reduced biological activity (Johannsen et al., 2008).

Under normal flow conditions, nitrate concentration then decreases due to assimilation and biomass production (Fig. 1). As a consequence, dual isotope values then negatively correlate with nitrate concentration (Montoya and McCarthy, 1995; Voss et al., 2006; Johannsen et al., 2008). In our study, $\delta^{15}$N-$NO_3^-$ and $\delta^{18}$O-$NO_3^-$ are negatively correlated with [$NO_3^-$] only after the SPM peak ($R^2$ of 0.897 and 0.816, respectively), also pinpointing reduced biological nitrate assimilation and showing that during flood conditions, assimilation by phytoplankton diminishes, probably due to high turbidity, short residence times, and decreased light availability (Voss et al., 2006; Deutsch et al., 2009).

From 14 June on, dropping discharge allows a recovery of phytoplankton, and rising oxygen concentrations indeed suggest that phytoplankton is recovering. In addition to the clear anticorrelation to [$NO_3^-$], dual nitrate isotope values further indicate rising phytoplankton activity. At the beginning of the flood, $\delta^{15}$N-$NO_3^-$ is not correlated with $\delta^{18}$O-$NO_3^-$, but after the SPM peak, both isotopes change almost in parallel along a slope of 0.82 ($R^2$ of 0.960). This is close to unity, which is typical of phytoplankton assimilation (Granger et al., 2004; Deutsch et al., 2009).





### 4.2 The role of nitrification

As outlined above, nitrate trends can be explained by assimilation and hydrographic properties. This is not quite the case for nitrite and ammonium concentrations during the flood. Generally, these nutrients do not accumulate in the water column during spring time. During the flood, ammonium and nitrite, the substrates for nitrification, are present

in unusually high concentrations but then decrease again, indicating active nitrification.

We regard SPM as the main source of remineralized ammonium, which in turn is then usually immediately assimilated (Dortch et al., 1991) or oxidized to nitrite (Mayer et al., 2001), which then is rapidly detoxified by oxidation to nitrate (Alonso and Camargo, 2006; Philips et al., 2002). Ammonium in the river initially has an isotope value of ~2‰, approximately 4‰ lighter than the SPM pool (Fig. 2c). If ammonium stems from remineralization,

this suggests a 4‰ fractionation during remineralization, as it was recently put forward by (Möbius, 2013). The author found that during remineralization in marine sediments, isotope enrichment with an isotope effect of about 2‰ occurs. Our value is slightly higher, but it seems plausible that organic material in the Elbe River most likely is more easily accessible than that of marine sediments and hence may be subject to more intense fractionation. This fractionation during remineralization also explains the depleted initial $\delta^{15}N$-$NH_4^+$ values we find, and it is in

accordance with (Schlarbaum et al., 2011), who found differences of up to 4‰ between dissolved organic nitrogen and suspended matter in the Elbe River. It is unlikely that ammonium in the water column derives from agricultural sources, because the positive charge of ammonium molecules tightly binds them to clay particles in soil, and elution with discharge generally does not occur (Mancino, 1983; Mancino and Troll, 1990).

When ammonium drops below the detection limit with decreasing discharge, nitrite remains above 3 µmol L$^{-1}$ (Fig.

2b). This succession of nitrite and ammonium concentration maxima can be taken as an indicator of successive nitrification acting as ammonium and nitrite sink, respectively (Meeder et al., 2012). Ammonium and nitrite accumulate at oxygen concentrations below 7 mg L$^{-1}$ (Fig. 3). As outlined above, phytoplankton activity at this time is low, and nitrification is the main ammonium sink. Ammonium-oxidizing bacteria (AOB) are active under low [$O_2$], but ammonium oxidation obviously cannot keep pace with remineralization, so that ammonium accumulates.

$\delta^{15}N$-$NH_4^+$ of the residual increases, because ammonium oxidation has a strong isotope effect of -14 to -38‰ (Delwiche and Steyn, 1970; Mariotti et al., 1981; Yoshida, 1988; Casciotti et al., 2003). Simultaneously, nitrite concentration peaks at low oxygen concentrations and decrease again, when oxygen concentration rise above 7 mg L$^{-1}$ (Bernet et al., 2001; Jianlong and Ning, 2004). $\delta^{15}N$-$NO_2^-$ are negatively correlated to nitrite concentration (fig. 2b, c). Altogether, we see a signal of coupled SPM remineralization and concomitant nitrification to newly produced

nitrate in the Elbe River at lowered [$O_2$].

Within 7 days, SPM is <14 mg L$^{-1}$, and ammonium concentration again falls below the detection limit due to complete oxidation of ammonium by AOB. If ammonium were mainly removed via nitrification, this decrease corresponds to an ammonium oxidation rate of about 0.5 µmol L$^{-1}$ d$^{-1}$, which is well within the range of water column nitrification rates for temperate river systems (Bianchi et al., 1994; Daims et al., 2015; van Kessel et al.,

2015). Nitrite concentration drops to typical spring/summer values (i.e. <1 µmol L$^{-1}$) after the flood event.

While we cannot trace the newly produced nitrate from nitrification into the much larger nitrate pool, the gradual change of nitrite concentration and isotope values provides the unique opportunity to for the first time calculate the isotope effect during nitrite oxidation in the Elbe River.





### 4.3 No inverse fractionation during nitrite oxidation

During the period with decreasing nitrite concentrations, we calculated the isotope effect of nitrite oxidation assuming closed-system Rayleigh fractionation (Eq. 2), because ammonium is not abundant in the water column during most of the sampling period, and hence input of new nitrite from ammonium oxidation is hypothetical. In a

closed-system, $\delta^{15}N\text{-}NO_2^-$-values should behave linear to $\ln([NO_2^-]/[NO_2^-_{initial}])$, and the slope of the regression line corresponds to the isotope effect (Mariotti et al., 1981; Scott et al., 2004). For the time span with decreasing nitrite concentration, we calculated an isotope effect of -9.3‰±0.6‰. (Fig. 4). To validate our results, we compared them to an open system approach and calculated an isotope fractionation factor based on an open system steady-state model (Sigman et al., 2009). Here, the fractionation factor is -10.6±1.0‰, not significantly different from the closed-

system approach, also suggesting conventional negative fractionation during nitrite oxidation. This means that during processing, i.e. nitrite oxidation in the water column, the remaining nitrite pool gets subsequently enriched in 15N. This result is surprising, because pure culture experiments with the marine nitrite oxidizing bacteria *Nitrococcus mobilis* (Casciotti, 2009; Buchwald and Casciotti, 2010), *Nitrobacter sp. Nb 355* and *Nitrospira marina* (Buchwald and Casciotti, 2010) suggested that nitrite oxidation exhibited rare inverse isotopic fractionation.

While some uncertainty naturally arises from the fact that we did not investigate pure cultures, but a biologically diverse setting like the Elbe River, we would like to investigate whether it is plausible that bacterial nitrite oxidation in natural environments rather follows "conventional" than inverse isotope fractionation. This would imply molecular differences on the enzyme level between naturally occurring and isolated nitrite oxidizing bacteria.

Fractionation factors can indeed vary greatly between different species (Casciotti et al., 2003). Nevertheless, the

inverse fractionation during nitrite oxidation is assumed to be based not on enzyme geometry, but on the stability of an intermediate during the oxidation process: Greater stability of – otherwise labile – intermediates containing $^{15}N$ facilitates further reaction and thus, nitrite oxidation. This transition state should be identical among nitrite oxidizing bacteria and should not depend on specific (or species-specific) enzyme equipment. Hence, we assume that an alternative, conventional isotope effect in natural assemblages of nitrite oxidizers is unlikely; the divergent isotope

effect in our study must have another reason.

The changing nitrite isotope concentration and isotope values are not caused by water column denitrification, because oxygen concentration is above 6 mg L$^{-1}$ (Lehmann et al., 2004) and sedimentary denitrification has little to no impact on water column nitrate isotopes (Brandes and Devol, 1997) and much less so for nitrite, there is no evidence of nitrite release to the water column. Sedimentary denitrification in the Elbe River generally is of minor

importance (25%) and nitrate assimilation by phytoplankton is the major sink (Deutsch et al., 2009). Nitrite is not present in atmospheric deposition (Beyn et al., 2014), so that we also exclude an effect of atmospheric deposition on nitrite concentration or isotope changes. Overall, changing nitrite isotope values must be due to nitrification.

During the flood event, discharge and turbidity are high, and remineralization of suspended matter is intense, as is evidenced by high ammonium and low dissolved oxygen concentrations during the flood. Shortly after the highest

discharge, $\delta^{15}N\text{-}NH_4^+$ is 2‰ (see above), but this value increases during the flood, reaching a maximum of 12‰. Over the course of the flood, oxidation of ammonium proceeds. Several studies suggest a high isotope fractionation factor during the oxidation of ammonium to nitrite, between -14 and -38‰ (Delwiche and Steyn, 1970; Mariotti et al., 1981; Casciotti et al., 2003). Such high fractionation explains the increase of $\delta^{15}N\text{-}NH_4^+$ to 12‰ (Fig. 2c).



This ammonium recycling also offers an alternative explanation for the aberrant isotope fractionation effect we calculated for nitrite oxidation. Ammonium concentrations are below the detection limit at the time of decreasing nitrite concentrations, but we speculate that rapid remineralization and oxidation of ammonium proceed. In this case, newly generated, isotopically enriched nitrite from ammonium oxidation would (a) prevent ammonium accumulation
and (b) camouflage the inverse isotope effect of nitrite oxidation, because the generation of new, of enriched nitrite exceeds nitrite turnover by nitrite oxidation.

However, the ammonium concentration is under the detection limit (0.5 µmol $L^{-1}$) when nitrite removal occurs - can we thus assume that ammonium oxidation is still active at such low concentrations?

Cryptic element cycling, with *in situ* substrate concentrations near the detection limit, has indeed been reported on
various occasions. In the OMZ off the Chilean coast (Canfield et al., 2010) propose that sulfate reduction and sulfide oxidation may long have been overlooked, due to the close coupling of these processes, that leaves no chemical evidence of their activity. Similarly, rate measurements also revealed the activity of anammox in OMZ water at no more than nanomolar concentration of ammonium (De Brabandere et al., 2013). If ammonium recycling is relevant even at such low concentrations, it seems plausible to assume that ammonium oxidation by nitrifying bacteria also
occurs in our setting, where concentrations below the detection limit may be as high as 0.5 µmol $L^{-1}$.

In such a case, ammonium will rapidly be recycled, and isotope signature of ammonium will be transferred directly to the nitrite pool, because limiting ammonium concentrations prevent any significant fractionation during oxidation (Mariotti et al., 1981). As a result, even if nitrite oxidation is coupled with inverse fractionation, the resulting bulk $\delta^{15}N\text{-}NO_2^-$ values will show an apparent enrichment due to "dilution" with nitrite derived from relatively heavy
ammonium.

In a back-of-the-envelope calculation, we tested this hypothesis. We assumed inverse fractionation with an isotope effect of +10‰ that acted on nitrite, assuming standard Rayleigh fractionation, and that nitrite oxidation and ammonium oxidation occurred at comparable rates. We found that the resulting isotope value of nitrite indeed showed conventional negative isotope fractionation of ~-8‰, close to our measurements, but only if we assumed that
the isotope value of ammonium increased with an isotope effect for ammonium oxidation of -35‰. This, barring considerable uncertainties regarding reaction rates and enrichment factors, this calculation nevertheless suggests that ammonium recycling can change the observed isotope effect of nitrite.

## 5 Conclusions

Our data suggest that nitrification is the main sink of ammonium and nitrite, respectively, in the Elbe River during
the flood, where we find intermediate ammonium and nitrite accumulation in the water column. While the main source of ammonium is remineralization of organic material, increasing isotope values over time indicate active microbial ammonium turnover and oxidation of ammonium to nitrite. The subsequent decrease of the nitrite concentration in the water column indicates nitrite oxidation, which surprisingly does not exhibit inverse fractionation, but follows conventional fractionation, with subsequent isotope enrichment of the nitrite substrate
during processing. For this removal, we calculated an isotope effect of -9.3‰. We hypothesize that the reason for this surprising isotope effect is a cryptic ammonium cycle, which quickly channels newly generated, isotopically




heavy nitrite into the nitrite pool. This continuous dilution masks the inverse enzymatic isotope effect of nitrite oxidation.

Our data demonstrate that results from pure culture experiments cannot easily be extrapolated to natural systems, because computed isotope effects depend not only on substrate concentration, but also are subject to a complex interplay of biogeochemical processes. While nitrogen isotopes are a powerful tool to unravel individual processes, divergent isotope effects and individual process rates need to be taken into account. The inverse isotope effect of nitrite oxidation adds more complexity to the isotope budget of the aquatic nitrogen cycle, but our data suggest that it might not be expressed in natural environments at all. If this is the case, it holds true not only for estuarine settings, but probably for any environment that shows nitrite accumulation in the water column, like oceanic OMZs, where nitrate and nitrite isotopes are frequently used to assess nitrogen dynamics.

## 6 Acknowledgement

We thank F. Langenberg and P. Martens from the University of Hamburg for the analysis of suspended matter samples. The Helmholtz Association is acknowledged for funding (VH-NG-721). B. Gaye is gratefully acknowledged for helpful comments on an earlier version of this manuscript. We also thank J. Kappenberg from the Helmholtz Centre Geesthacht for providing the hydrological archive data sets.

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

**8 Figures**

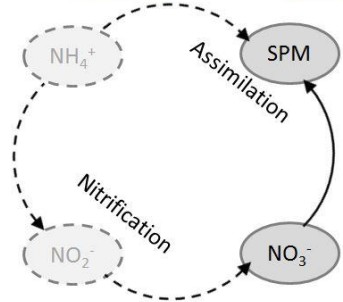

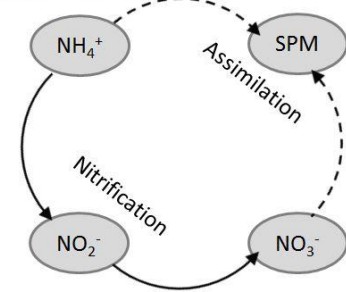

Figure 1



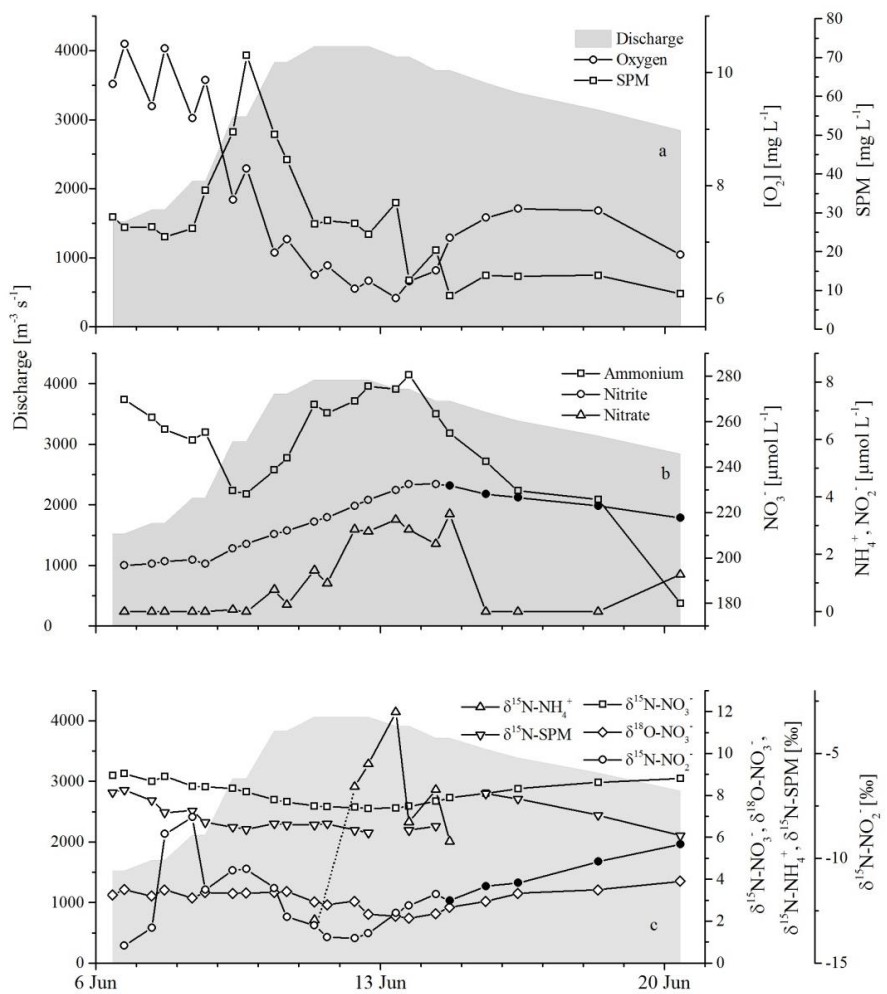

Figure 2a, 2b, 2c





Figure 3

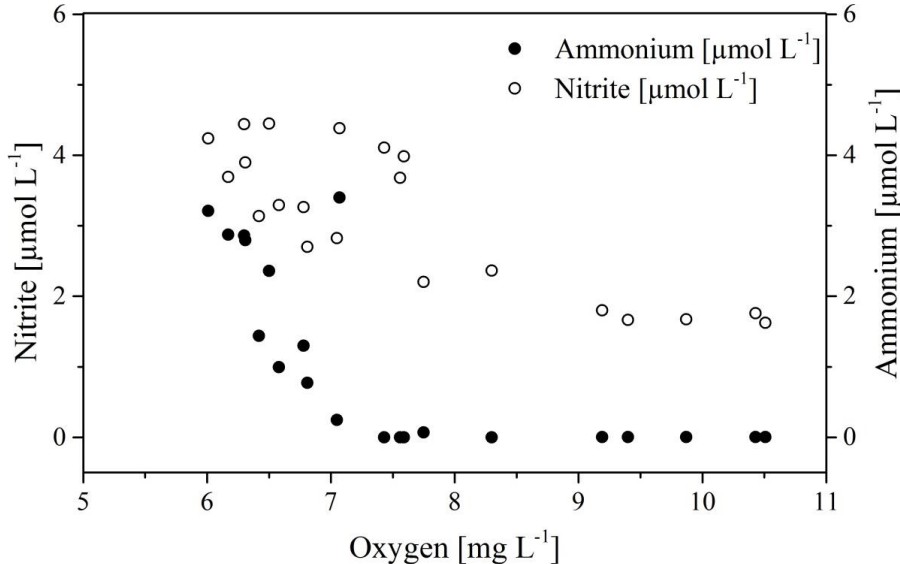

Figure 4





**9 Figure Captions**

Figure 1 Sketch of summer condition at normal discharge where light colouring indicates concentration below the detection limit compared to the flood condition. Active nitrification is observed during the flood, but assimilation diminished shown as dashed lines.

5 Figure 2a Discharge, dissolved oxygen concentration, and SPM concentration of the Elbe River water samples during the flood from 6 to 20 June 2013.

Figure 2b Ammonium, nitrite, and nitrate concentrations in the Elbe River in the course of the flood. Calculation of the fractionation factor is based on filled dots data.

Figure 2c Ammonium, nitrite, nitrate, and SPM isotope values in the course of the flood. Filled symbols indicate data
10 that were used for calculation of the fractionation factor.

Figure 3 Ammonium and nitrite concentrations increase with decreasing dissolved oxygen concentration.

Figure 4 Rayleigh plot for nitrite oxidation during the Elbe flood. Data points correspond to the filled symbols in figure 2b and 2c. Dotted line indicates least square fit with a slope of -9.32 indicating $^{15}\varepsilon$, and $R^2$ is 0.98.