# Peer review of "Nitrite consumption and associated isotope changes during a river flood event"

_Biogeosciences, 2016_

## Referee Comment (RC1) · Anonymous Referee #1 · 23 Mar 2016

**"Nitrification and Nitrite Isotope Fractionation as a Case Study in a major European River",
by Juliane Jacob, Tina Sanders, and Kirstin Dähnke**

In this submission, the authors present concentrations and N (and O) isotope compositions of N-species during a flood event in the Elbe River in summer 2013. The dataset includes nitrate, nitrite, ammonium and SPM (concentrations and isotope compositions) data, which were analyzed simultaneously and in high resolution (up to twice a day), although unfortunately only for a short period of time (14 days). The presentation of this dataset in the main Fig. 2 is good, however, I have some major concerns regarding the discussion and main conclusions (see also general comments below). The biggest problem, in my opinion, is that neither independent rate measurements nor convincing clues on what processes are active/negligible exist. This makes data interpretation difficult. Instead of trying to deduce isotope fractionation factors with a Rayleigh approach (which I think is not justified in this system), I would suggest to use a simple reaction-model with reaction rate constants as tunable parameters (and assuming literature fractionation factors). What processes need to be active and at what rates in order to reproduce all of your concentration and isotope data? Do these predicted rates make sense? Or do you see any reason why your underlying assumptions (e.g., literature fractionation factors) could be different in your system?

Datasets like the one presented here are scarce but interesting. I therefore see potential for its publication in *Biogeosciences*, but only after addressing the major issues raised here.

**General comments**

I have a few general concerns with the data interpretation. First of all, it is not clear to me why the authors conclude that all nitrite must be oxidized to nitrate. Why can we exclude nitrite assimilation into biomass (while nitrate assimilation was the main argument for nitrate removal…)? And why was dissimilatory nitrite reduction disregarded? I understand that the water column is oxic, and denitrification unlikely, but what about sedimentary denitrification and nitrite reduction? Apparently 25% (which is not negligible!) of nitrate loads to the Elbe River is lost through denitrification (page 8, line 30).

Second, I don't think a Rayleigh model is of much use here, where there is simultaneous nitrite consumption and production, mixing and dilution. If the authors decide to keep such a calculation of apparent fractionation factors in the manuscript, the results should be discussed much more carefully and adequately. To conclude that in this environment we observe a normal isotope effect associated with nitrite oxidation is wrong: First, the calculation of the isotope effect is biased, second we don't know if nitrite oxidation to nitrate is the only nitrite removing process (see comment above). On the other hand, if an apparent fractionation factor is calculated for nitrite consumption, why not for nitrate removal as well (which the authors claim is through assimilation only)? Also, the $^{18}O$ to $^{15}N$ enrichment during nitrate reduction (assimilatory or dissimilatory) should be independent of mixing/dilution and could be more diagnostic of the actual process affecting nitrate dynamics. The authors calculate an $^{18}O$ to $^{15}N$ enrichment ratio of 0.82 (page 6, line 32), which is not characteristic for nitrate assimilation.

**Specific questions and comments**

The title is rather weak. What does "nitrification as a case study" mean? I would change the title to something like "Nitrification and nitrite isotope composition after an exceptional flood in the Elbe River". You could also change "a major European River" to "the Elbe River".

Page 1:

Line 8. Why do we care about a nitrate source from $NH_4^+$? From an environmental perspective the source of total fixed N is important to know.

Line 9. What do you mean with recycling processes? DNRA?

Lines 10-12. Do you mean the isotope effects of ammonium oxidation to nitrite and nitrite oxidation to nitrate? Why do we care about isotope effects associated with nitrite reduction/oxidation if no nitrite accumulates? The isotope effect of ammonium oxidation should be easier to predict if no nitrite accumulates, not more difficult.

Line 16. Delete "the" before concert.

Line 16. Define SPM.

Line 18. What do you mean by "internal recycling processes"? DNRA?

Line 18. State that xy% of reactive N in the water column was present as nitrate, which leached from the catchment area. This also contradicts the first sentence of the abstract, which says that nitrification is "an important nitrate source" in the water column.

Line 19. The maximum ammonium concentration stated here does not agree with Fig. 2b. After reading all the paper I think there is a mistake in the legend in Figure 2b. The symbols for ammonium and nitrate concentrations are mixed up?

Line 20. Write here on what basis you conclude that the increase of ammonium and nitrite concentrations is due to remineralization and nitrification.

Line 25. What is the justification then to use a Rayleigh model?

Line 26. I think this statement is not justified. The data are not a first approximation of the isotope effect of nitrite oxidation in a natural environment. First, as stated one line above, we have co-occurring nitrite production, and second, we have not seen a proof that nitrite is indeed oxidized (and not assimilated/reduced).

Line 33. Write "dissolved inorganic nitrogen (DIN)".

Line 34. What about Nr inputs? Ammonium input has been reduced, so now the same amount of Nr is introduced to the system as nitrate?

Page 2:

Line 4. Sentence starting with "nevertheless" makes no sense. Nitrate is part of DIN.

Line 15. The fractionation factor can be calculated differently, not only approximated using a Rayleigh distillation model.

Line 16. (Half-)sentence makes no sense. What are the "individual uptake processes"?

Line 26. What do you mean by "relatively healthy"?

Page 3:

Line 16. How were samples taken? From the shore? From a boat?

Line 21. Define SPM.

Line 22-24. Please give a detection limit and analytic precision for nitrate, nitrite, and ammonium concentration measurements.

Line 26. "Dual nitrate plus nitrite…"

Page 4:

Line 1. How was ammonium converted to nitrite? Does this mean that the isotope measurements include the naturally abundant nitrite? Why was the produced nitrite not reduced to $N_2O$ with the bacterial method as stated above, but with sodium azide instead?

Line 9. Delete ", which both by definition have a delta value of 0‰". This is obvious.

Line 18. What are the "additional standards"?

Lines 19-23. See my comments above. State here under what conditions a Rayleigh model can be used. Why do you think these conditions are met in your system? What could such an apparent fractionation factor tell us, what certainly not. What are the limitations. It is too simplistic (even wrong) to just say that a fractionation can be calculated this way in your system.

Page 5:

Line 6. I guess the cited 2015 is a mistake, if the here presented data are from 2013?

Line 12. In the abstract is was a maximum of 4.5 µmol $L^{-1}$. Overall, there is only a 1.1 µmol $L^{-1}$ decrease over 6 days. What is the precision of your measurements?

Line 15. Logic is not clear. First nitrite is discussed, then ammonium, then nitrite again.

Line 16. What is this limit of 2.7 µmol $L^{-1}$?

Line 16. We see an increase in nitrite concentrations and we also see a decrease in oxygen concentrations. The nitrite concentration increase is not necessarily due to an oxygen concentration decrease.

Line 27. Here and throughout the manuscript: define "flood" and use it consistently. Or what does "before the flood" mean here?

Line 27. I don't see the nitrite concentration increase in the figure. Can you give numbers?

Lines 28-29. The isotope minimum does not correspond to the concentration maximum. Why?

Line 30. See comment above and avoid saying "calculated" fractionation factor, as if this would be a real/robust value. With a Rayleigh model you can at best approximate a fractionation between a substrate and a product, and only under certain assumptions (no co-occurring production, product continuously removed from a well-mixed system, etc.)

Line 35. Is the legend in Fig. 2b mixed up? Otherwise this statement makes no sense.

Line 36. Explain why you compare $\delta^{15}N\text{-}NH_4^+$ to [SPM]. Also, I do see some zic-zac correlation between the two. The second half of the sentence ", but not so much for $\delta^{15}N\text{-}SPM$" makes no sense.

Page 6:

Line 2. What means "initial"? Your first measurement?

Line 8. Legend in Fig. 2b is mixed up. Also give here the actual nitrate concentrations.

Line 10. The sentence makes no sense. Or what does "an increase in SPM peak" mean?

Line 10. Sentence starting with "both…" needs to be rephrased.

Line 13. How do you explain the time shift between the [SPM] and [$NO_3^-$] peak? If the same mechanism is causing the peak (leaking from agricultural soils) I would expect a simultaneous peak?

Lines 13-14. I don't understand the point made here. Can you elaborate more on why you expect a $\delta^{15}N\text{-}SPM$ decrease from 7.8 to 6.2‰? And why would along this argumentation $\delta^{15}N\text{-}SPM$ increase after 13 June and then decrease again after 16 June?

Line 21. "reduced biological activity" relative to what? To summer, when we observe average $\delta^{15}N\text{-}NO_3^-$ values of … ‰?

Line 22. What "then"? in summer?

Line 23. Add a reference to where it was shown that nitrate concentrations decrease due to assimilation only (and no denitrification).

Line 29. Rising oxygen concentrations to above saturation? Otherwise it would be more interesting to know what caused the [$O_2$] decrease before the system is ventilated again.

Line 30. Can you show here this correlation?

Line 32. Can you add a plot showing the $\delta^{18}O$- to $\delta^{15}N\text{-}NO_3^-$ increase?

Line 33. No! Exactly the opposite is the case. 0.82 is significantly different from 1.00 ± 0.01, which could have been indicative of assimilation (Granger et al. 2004). The lower value of 0.82 is interesting though. Can you elaborate on the reason for the observed $^{18}O$ to $^{15}N$ enrichment ratio?

Page 7:

Line 2. I am not convinced at all. You did not prove that nitrate concentrations decreased due to assimilation only. In contrary, the $^{18}O$ to $^{15}N$ enrichment ratio seems to be indicative of other processes.

Line 2. What do you mean by "explained by hydrographic properties"?

Line 3. What is "not quite the case"? Why can nitrite and ammonium concentrations not be explained by assimilation and "hydrographic properties"?

Lines 4-5. There are lots of conclusions in this sentence without any argumentation, underlying data, or proofs. Ammonium and nitrite are also substrates for assimilation and anammox, and nitrite can be reduced to $N_2O$, $N_2$, or ammonium. A decrease in ammonium and nitrite concentrations is hence not a proof for nitrification.

Line 6. Start here with a short discussion of where the $NH_4^+$ can come from, instead of just claiming that all of it is originating from SPM remineralization.

Line 8. With "initially" you mean your first measurement?

Lines 14-15. Sentence not needed. What do you mean by "this also explains"?

Line 14. "…between $\delta^{15}N$ of dissolved…"

Lines 16-18. Move this discussion up. (see comment above, line 6). Also, SPM is leaching, so $NH_4^+$ could be attached to it and thus leaching as well?

Line 22. Why are $O_2$ concentrations "low"?

Lines 22-23. But phytoplankton activity is not 0, right? Based on what are you so convinced that nitrification is the ammonium sink?

Line 23. Why do you need to say that AOB are active under low $[O_2]$? What do you mean by low $[O_2]$? 6 mg/l is not really low.

Line 24. Why can nitrification "obviously" not keep up with remineralization? Why is this obvious?

Line 27. Define "low oxygen concentration".

Line 28. Not really. Only the last five data points (out of 21) show a negative correlation.

Lines29-30. Where do you see the "newly produced nitrate"? If nitrate is produced it contradicts the first sentence of this paragraph, which says that nitrate dynamics are controlled by assimilation and "hydrographic properties". And again, why do you completely rule out any ammonium assimilation? Why would only nitrate by assimilated and not ammonium?

Line 30. What causes the lowered $[O_2]$?

Line 31. Within 7 days of or after what?

Lines 31-32. Again, this is an overstatement. You have not proven that all ammonium is oxidized to nitrate and even less so by AOB.

Line 33. Where is that 0.5 $\mu$mol $L^{-1}$ $day^{-1}$ coming from? I see a sharp decrease in $[NH_4^+]$ of about 3 $\mu$mol $L^{-1}$ $day^{-1}$ on 15 June 2013.

Line 35. Where is that drop in nitrite concentrations after the flood? In Fig. 2b $[NO_2^-]$ remains at >3 $\mu$mol $L^{-1}$.

Page 8:

Line 1. Title is misleading (wrong) and contradicts your conclusion from the last paragraph on page 9.

Lines 3-4. No input of new nitrite? This directly contradicts your conclusions e.g., on page 9, line3.

Line 5. How to you consider the surface water of a river a "closed system"?

Line 8. What was in steady-state during the flood?

Line 10-11. Again, I don't believe that all nitrite is oxidized to nitrate in the water column. What about dilution of the signal with other water masses? Nitrite assimilation? Sedimentary denitrification? (not to mention other processes such as anammox, DNRA in the sediments).

Lines 15-25. This is pretty obvious and could be deleted.

Line 27. Lehmann et al. (2004) do not show that denitrification does not take place in the Elbe River, nor that $O_2$ concentrations were >6mg/L.

Lines 26-29. Sentence makes no sense and needs to be rephrased.

Lines 27-28. Sedimentary denitrification can be the reason for low apparent isotope effects in the water column.

Lines 29-30. 25% of what? If 25% of nitrate is reduced to $N_2$ (or $N_2O$) it is not negligible!

Line 32. I don't agree. (see comments above).

Page 9:

Lines 21-27. Generally, it is very difficult to interpret the isotope signatures presented in this study without any idea about reaction rates. Could you, instead of assuming a Rayleigh model, develop a simple reaction model, which allows you to reproduce all of your measured data (nitrate, nitrite, ammonium, and SPM concentration and isotope data (at least after 14 June)), based on reaction rate constants as tunable parameters? This would give you very important information on what processes could have been active during the flood. I feel this paper could be improved significantly with such a model.

Page 10:

Line 8. What do you mean by "not expressed". It is expressed but overprinted by other effects?

Figures:

Figure 1. Caption. Define "flood". I thought you argued that assimilation was the one and only nitrate sink?

Figure 2b. Legend/symbols got mixed up (see comments above). How do you explain the sharp drop in ammonium concentration on 15 June 2013?

2c. What is the dashed line? Why are there gaps in the $\delta^{15}$N-SPM profile?

Figure 3. I don't think this figure is of much use. Or explain in the caption (and main text) what we are supposed to learn from it.

Figure 4. Where is $\Delta\delta^{15}$N-NO$_2^-$ defined? Write in caption what [NO$_2^-$]$_{initial}$ is.

Generally: check for typos and text formatting throughout the manuscript and specifically on pages 4-7.

---

## Referee Comment (RC2) · Anonymous Referee #2 · 28 Mar 2016

Review of "Nitrification and nitrite isotope fractionation as a case study in a major European River" by Jacob et al.

Summary

The authors present a study that depicts the evolution of nitrogen cycling dynamics during a large flood of the Elba River in Germany during June 2013. Using a combination of both concentrations and isotopes of nitrate, nitrite, ammonium and particulate nitrogen, the authors aim to understand the nature of the biogeochemical processes responsible for these changes and what can be learned about the controls on them – in this case under flood conditions – but perhaps in rivers in general. In particular, the authors argue that light limitation by the elevated suspended load during the flood acts to limit photosynthetic uptake by phytoplankton in the river – offering a perspective on

nitrogen cycling largely in the absence of assimilatory processes. Based on this and other assumptions, the authors conclude that nitrification is driving the majority of the observed patterns and then use the isotope data to make estimates of nitrogen isotope fractionation for nitrification.

General Comments Overall – this is an interesting dataset with relevance to understanding the nature and magnitude of nitrogen cycling processes in large rivers. To my knowledge these type of datasets are fairly novel (especially the nitrite isotope analyses) – and certainly reflect a complex array biological and hydrological processes. To the degree that a mechanistic understanding of the processes reflected in the concentrations and isotopic compositions can be refined, this study represents a novel step forward in the development of such isotopic tools. However, my primary concern is that rivers are inherently dynamic, non-steady-state systems – and that it may be difficult or impractical for the authors to isolate a single biogeochemical process within this physically complex and hydrologic system (confounded by factors such as dilution?, source-mixing?, hyporheic flow?). Studies of hydro-biogeochemical processes are notoriously complex – in particular over the course of a large episodic event – in which the proportions of primary flow paths, for example, (e.g., soil water, shallow groundwater, deep groundwater, hyporheic exchange, surface runoff, etc.) may also be changing over time. In fact, the authors acknowledge that the increase in the nitrate and SPM concentrations on the rising limb and crest of the flood reflect changes in sources of watershed inputs (e.g., P6 L10-14).

Since samples were collected at only one point on the river – the perspective for quantitatively evaluating N cycling processes within the river is somewhat limited. As presented, it is hard to discern to what degree hydrologic changes might account for the observed changes in isotopes and concentrations. Indeed this perspective embodies a classic sampling perspective for riverine/hydrologic studies: Eulerian (fixed point or volume) versus Langrangian (fixed water parcel). The authors' study is intrinsically Eulerian observing a defined volume (control volume) occupying a fixed point (or box) in

space. However, since many (most?) of the features being observed by the authors' measurements may indeed be related to changes in the hydrologic and geochemical inputs to this volume – it becomes virtually impossible to assign the biogeochemical changes observed to processes occurring within the box (as the authors have attempted to do). Rather, the authors' questions (how much nitrification occurs and/or plays a role in the isotopes, for example) would be better addressed by a Lagrangian approach in which a parcel of water is tracked down river over – such that changes in the inputs to the system can be more or less neglected and the changes in the nitrogen content and isotopic composition can be directly related to 'in-river' processes.

Clearly, one cannot expect the authors to repeat the study of this extreme flood event. Can any other conservative tracers of flow ($\delta 18O$ water, bromide, chloride, major ions, etc.) be measured to help constrain water (and N) sources during the flood hydrograph? I wonder whether a simple box model could be constructed, in which, one might solve for varying rates of nitrogen cycling processes required in order to fit the observed data – and something then be learned about the operation of varying processes under such flow conditions? If we can make the assumption (based on conservative tracers?) that the water and N sources are conservative (and constant?) and that all biogeochemical reactions happen in the river (including hyporheic exchange?) – and make estimates of them using conservative tracers - then perhaps variations of the observed compositions could be used to constrain rates of those biogeochemical processes.

Another thought is that the overall discussion might benefit from a re-focusing around the "fate of nitrite" and "nitrite-consumption processes" – rather than solely on nitrite oxidation (including nitrite assimilation, oxidation or denitrification). Ultimately the authors conclude that multiple processes are in play here. Thus, while the authors may not be able to nail down one specific process – perhaps they could make reasonably well-constrained estimates of the rates of multiple processes.

Additionally, I don't think a closed system Rayleigh model can be justified here. In

general - use of the Rayleigh fractionation model implicitly assumes that only one, uni-directional process is occurring. In addition to nitrite oxidation, however, by the authors' own argument – the nitrite isotope data likely reflect at least one other processes (ammonia oxidation) – thereby invalidating the use of a Rayleigh model for estimating the isotope effect of a single process. While it is possible that the decrease in NO2- concentration is caused by a river-hosted biological (e.g., fractionating) process – leading to the observed increase in N isotopic composition, can it be demonstrated that the nitrite concentrations are not the product of low levels of NO3- reduction occurring in sediments/hyporheic exchange/groundwater? In fact, the nitrite concentrations vary in a smooth fashion (in contrast with the NH4+ concentrations, for example) – which to me might suggest more of a hydrologic control on their dynamics.

After much confusion - I think that the Figure 2B legend is wrong.

The decrease in nitrate concentrations on the falling limb of the flood are explained by phytoplankton assimilation – why could this not also possibly explain the concomitant los of nitrite and the positive isotope excursion?

Specific Comments

P1 L9: 'bulk isotope effect of nitrification' is not clear and should be defined.

P1 L11: 'divergent' is unclear

P1 L16: In concert with...

P1 L19: ...from the catchment area.

P1 L22: I'm not convinced that you can conclude the changes in isotopes are the result of nitrite oxidation only. You should state that this is the 'apparent' isotope effect of nitrite consumption (although this may also not be valid as calculated – e.g., violation of Rayleigh model assumptions).

P1 L30: ... has increased 20-fold...

P2 L7: Or, the nitrate can be simply exported from the watershed.

P2: I think the imperative for understanding nitrification in riverine systems should be better justified – perhaps in terms of its frequent coupling to N loss processes (anammox and denitrification) and ecosystem services.

P2 L13: Not just nitrogen uptake – but any enzymatically catalyzed nitrogen transformation process.

P2 L14: This is somewhat of a colloquial expression – and should be restated to reflect that enzymatically catalyzed processes occurring slightly faster for lighter isotopes that heavy isotopes.

P2 L15: The Rayleigh model explicitly requires the assumption of a unidirectional process and no replenishment of the reactant pool. It's not clear that this can be assumed.

P2 L18: obstacle to what?

P2 L21: . . .and the remaining nitrite. . .

P2 L26: what is meant here by the term 'healthy?'

P2 L37: Phytoplankton are light dependent. . .

P3 L8: . . .the second largest river discharging. . .

P3 L25: Isotope analyses

P5 L4: Previous studies have found. . .

P5 L8: Either present data as singular or plural – not both. . . . nitrate concentrations were. . . Nitrite concentrations were <1.2. . . and ammonium concentrations were below the detection limit. . .

P5 L28: Not sure I would say that the nitrite concentrations rose 'quickly' – they seem to evolve more gradually in fact.

P5 L30: For reasons discussed above, I think it should be stressed here that this is an 'apparent' fractionation factor.

P6 L2: Remove "it is interesting, however" – opinions don't generally belong in a results section.

P6 L 20: Although as noted later – the relatively large nitrate pool is far more resistant to isotopic perturbations by biogeochemical processes.

P6 L25: While this may be true – the watershed flooding potentially may have also introduced a nitrate source having a slightly different isotopic composition.

P6 L29: As the phytoplankton are recovering – couldn't they be assimilating nitrite?

P6 L32: This $\delta$15N vs $\delta$18O slope is actually much lower than that observed by Granger and colleagues. Could this be indicative of nitrification?

P7 L3: Why can't phytoplankton be playing role in assimilation of nitrite and/or ammonium?

P7 L23: But phytoplankton activity was specifically invoked as explaining an increase in N and O isotopes of nitrate and contributing to a drawdown of $\sim$100uM nitrate. Thus, it seems hard to discount phytoplankton activity for drawdown of $\sim$1uM nitrite and $\sim$3uM NH4+.

P8 L10: ... suggesting conventional normal (as opposed to inverse) fractionation during...

P8 L29-30: The contribution of 25% sedimentary denitrification actually seems substantial – and therefore hard to rule out. Also – can the same conclusions of Deutsh et al., 2009 be drawn for the extreme flood conditions of this study? Couldn't flooding act to increase hyporheic exchange?

P8 L35: As conceptualized by the authors, since the $\delta$15N of the NH4+ is $\sim$+2permil to begin with ($\sim$4permil lower than the SPM $\delta$15N) – the $\delta$15N of the NO2- produced (in a

closed system) would follow the accumulated product equation – and under conditions where NH4+ was being completely oxidized to NO2- - the newly produced NO2- would have a $\delta$15N of +2permil. In developing the argument about the contribution of heavy nitrite from ammonia oxidation, the authors should be careful to explain how this new nitrite composition will evolve in step with the degree of NH4+ consumption. Initially the new nitrite will have a $\delta$15N even lower than the existing nitrite, while as NH4+ is consumed – the $\delta$15N of the newly produced nitrite will approach the original $\delta$15N of the NH4+ ($\sim$+2‰. This value of +2‰ is actually not the 'isotopically enriched' nitrite that seems to be invoked here by the authors. Later on P9 L18, the authors explain how the complete consumption of NH4+ would quantitatively transfer the $\delta$15N value of the NH4+ pool into the nitrite pool – yet it is not clear whether the authors are using the evolving NH4+ pool as a closed system – or simply invoking the instantaneous product equation at each step. Notably – these values and mass balance estimates will play importantly into their 'back-of-the-envelope' calculations.

P9 L21: I think the discussion could be clarified if these calculations were explained in more detail.

P9 L36: I don't think there is any sort of cryptic ammonium cycle occurring here. More likely, it seems that the authors are just witnessing more 'conventional' N cycling processes (e.g., remineralization, nitrification, assimilation, etc.) from the perspective of nitrite isotopes for the first time in a river.

Figure 2b: Caption is wrong?

Figure 4b: this should be labeled as 'apparent isotope effect for nitrite consumption' (not nitrite oxidation). As articulated by the authors in the discussion, I don't think you can tie these isotope changes to a single process.

---

## Referee Comment (RC3) · Anonymous Referee #3 · 6 Apr 2016

Overall, the data presented in the study are novel, particularly since there exist vanishingly very few measurements of nitrite isotopes, in tandem with ammonium and nitrate isotopes in any environment.

The interpretation of these results yields some constraints, but is otherwise wanting, for two overarching reasons: (1) The system is complex, and the data at hand are insufficient to resolve inherent dynamics and (2) interpretation of the isotope data is coarse, relying on over-simplifying assumptions.

(1) Evident when considering first of two main conclusions cited in the abstract

"We found that in the water column, ammonium and nitrite derived from internal recycling processes, whereas nitrate mainly leached from catchment area."

[Figure]

While this seems like a reasonable conclusion, I cannot decipher how the authors' data and their interpretation yield these conclusions.

For the sake of argument, could the ammonium and nitrite not be imported from the catchment, from internal cycling therein (in soil)? Which aspects of the isotope data enable partitioning of processes that happened in situ vs. the catchment? Does it even matter?

I think the general arguments/assumptions as to the origin and fate of ammonium, produced by recycling and consumed by nitrification are reasonable. Nevertheless, one could argue for some role of ammonium assimilation. Nevertheless, assuming negligible assimilation, could the authors not generate plausible scenarios of nitrite production/oxidation and associated isotope effects that could constrain the relative fluxes, given the measured isotope composition of ammonium and nitrite? I realize the range of solutions may be too broad, but perhaps some scenarios could be ruled out with such an exercise.

(2) The second conclusion stated in the abstract is facile and could be construed as misleading:

"Our data are a first approximation of the isotope effect of nitrite oxidation in natural environments and highlight that pure culture results cannot readily be extrapolated to natural microbial assemblages or water bodies."

The isotope composition of nitrite in the environment is implicitly the result of multiple co-incident reactions, each of which is associated with an isotope effect. It's self-evident that a single Rayleigh fit to NO2 consumption will not describe a single uni-directional reaction on said NO2, which does not mean that culture results cannot be extrapolated to the environment. What an odd conclusion! I urge the authors to refine this conclusion so as to appear less incongruous.

---

## Author Comment (AC1) · 27 May 2016

**Comments to „Nitrification and Nitrite Isotope Fractionation as a Case Study in a major European River" by Juliane Jacob et al. (2016)**

Referee comments in Times, **author responses in Arial**

**The authors have provided the information in the revised manuscript as requested by the reviewer. We thank the anonymous reviewers for their suggestions and the valuable concerns. In parts, the reviewers address similar issues, we decided to address these points jointly in more detail.**

**One concern was that no independent rate measurements and clues on what processes are active/negligible were done and that our assumption that nitrite oxidation was the main nitrite sink might not be valid.**

**With regards to this subject, we reconsidered our data, and we agree that our focus might have been too narrow to account for potential nitrite sinks. Accordingly, we have rephrased "nitrite oxidation" to "nitrite removal", and added a section about other potential sinks, like riparian denitrification, nitrite assimilation by phytoplankton, dilution and source-mixing.**

**Of these four potential processes, we assume that nitrite assimilation as a sink is of lesser importance. Even though the possibility of nitrite assimilation by phytoplankton is commonly accepted (Collos, 1998), it is energetically expensive because phytoplankton would have to reduce four electrons for every molecule of nitrite to assimilate nitrite. Furthermore, this reduction of nitrate to nitrite usually happens within the cell in the cytoplasm and the chloroplast, respectively. A direct assimilation of nitrite requires active transport through the chloroplast membrane and needs additional energy (Lomas and Lipschultz, 2006), making this process unfavorable in the presence of ample nitrate.**

**This process would not bias our isotope calculation, because nitrite assimilation in a pure culture was associated to a very small fractionation factor of -0.7 to +1.6‰ (Wada and Hattori, 1978), and thus would only have a minor influence on the isotope signature in the river.**

**Regarding denitrification, our initial assumption was that it would be negligible in the water column, because the oxygen concentration is above 6 mg $L^{-1}$, and that sedimentary denitrification, while potentially quantitatively important, has little to no impact on isotope values of the water column nitrate pool (Brandes and Devol, 1997; Mariotti et al., 1988; Mariotti et al., 1982). However, in the revised version, we consider riparian denitrification as a**

nitrite sink, which can have a notable apparent isotope effect (Mengis et al., 1999; Sebilo et al., 2003).

Dilution with water masses containing lower nitrite concentrations is unlikely because of the changing nitrite and nitrate isotope values. Source-mixing has also not been taken into account because nitrite is generally not abundant in the catchment and is immediately removed due to its toxicity (page 7, line 7).

We suggest the different shaped graphs of ammonium and nitrite concentrations and isotopes are not only influenced by hydrology, but more by biology. AOB and NOB have a different behavior/sensitivity to surface irradiance (Horrigan et al., 1981). NOB are more light sensitive (Olson, 1981) and poorly recover from photoinhibition (Guerrero and Jones, 1996). This could be a reason why nitrite can accumulate and the variations in concentrations and isotope values are less pronounced.

We did indeed not present rate measurements, however, we conducted incubation experiments to determine ammonium oxidation and nitrite oxidation rates over an annual cycle in 2012. We find nitrification rates of 1 to 14 $\mu$mol L$^{-1}$ d$^{-1}$ in winter and summer, respectively. However, due to time constraints, these measurements were not done during the flood event. In any case, such rate measurements can only serve as a proof that nitrification is active, because our sampling scheme does not really contain a temporal component, and rates cannot be connected to the isotope changes we see.

Another concern was the calculation of the fractionation factor of nitrite removal, which was based on Rayleigh closed-system equations. In the original manuscript, we decided to use this assumption, because ammonium concentrations are below the detection limit and from this perspective nitrite is the substrate being consumed. However, we reconsidered this and agree with the reviewers that this assumption is not valid in our case, as we also discussed in the original submission when we evaluated ammonium production. Consequently, we replaced the Rayleigh calculation with an open-system assumption (Sigman et al., 2009) in the revised manuscript. Using this approach, we calculated an apparent isotope effect of -10.0±0.1‰, which is still conventional.

Both reviewers suggested that the use of a simple box model or simple reaction model should be constructed to assess rates and processes occurring in the river. We took this into account and intensively discussed modeling options with two colleagues experienced in isotope modeling and in nutrient modeling in the Elbe. Our idea was to include isotopes in a biogeochemical model previously published by Friedhelm Schröder, who intensely studied the Elbe River in the 1980s and 1990s (Schroeder, 1997). However, both colleagues agreed that a model is nearly impossible to build

based on one sampling station only; because incoming and outgoing concentrations are basically unknown and no mass balance can be set up.

In response to the reviewers' suggestion, we decided to try what we considered the next best option. As nitrite trends during the flood are smooth and isotope changes follow a linear pattern, we assume that the ratio of nitrite processing pathways is constant, even though we cannot quantify rates. The source signal is the isotope value of the maximum nitrite concentration, and then calculated different scenarios with varying rates of nitrite oxidation, ammonium oxidation and denitrification to reproduce our measured data, assuming isotope effects from the literature.

These fractionation factors vary depending on involved microorganisms and environment (Buchwald and Casciotti, 2010; Casciotti, 2009; Casciotti et al., 2003; Delwiche and Steyn, 1970; Mariotti et al., 1981; Santoro and Casciotti, 2011; Yoshida, 1988), but within a range that appeared plausible, we varied these effects and corresponding rates. One plausible scenario is that we see a mixed signal of riparian denitrification and nitrite oxidation, with a constant replenishment of the ammonium pool from suspended matter. We will discuss these calculations in a revised version.

Anonymous reviewer #1:

**Reviewer #1: We would like to thank the anonymous reviewer for the detailed evaluation of our manuscript and for the suggestions for improvement. This reviewer had some general concerns regarding (1) the fact that we addressed nitrite oxidation as single nitrite sink, (2) the use of the Rayleigh closed-system equation to calculate the isotope effect of nitrite consumption and (3) suggests the use of a reaction-rate model to constrain relevant nitrite turnover pathways.**

**These issues are also brought up by Reviewer #2, and we addressed them jointly in a response addressing both reviewers (see above).**

I have a few general concerns with the data interpretation. First of all, it is not clear to me why the authors conclude that all nitrite must be oxidized to nitrate. Why can we exclude nitrite assimilation into biomass (while nitrate assimilation was the main argument for nitrate removal…)? And why was dissimilatory nitrite reduction disregarded? I understand that the water column is oxic, and denitrification unlikely, but what about sedimentary denitrification and nitrite reduction? Apparently 25% (which is not negligible!) of nitrate loads to the Elbe River is lost through denitrification (page 8, line 30).

**As we outlined above, we agree that our focus on nitrite oxidation only might have been too narrow. While we do have (as yet unpublished) independent rate measurements that support the occurrence of nitrite oxidation, these cannot be extrapolated to the flood situation. Regarding the role of denitrification, we now consider it as a potential turnover pathway in a revised version of the manuscript. However, we would like to point out that the 25% estimate of denitrification in the Elbe was a maximum estimate by (Deutsch et al., 2009).**

**With regards to DNRA, we do not consider it separately in the manuscript for the following reasons: First, we now compute a joint fractionation factor of nitrite sinks (assuming that denitrification and nitrite oxidation are the most important ones). The insertion of yet another sink will not significantly change our conclusions. Second, surprisingly enough, there seem to be no assessments of the isotope effect of DNRA, making it difficult to include in our nitrite sink calculation. Third, most importantly, we are not aware of rate measurements of DNRA in the Elbe, but (Burgin and Hamilton, 2007) assume that its contribution to nitrate removal in rivers is relatively low. This hypothesis is supported by a recent doctoral thesis (In-stream nitrogen retention in a large nitrogen rich river: estimates from open-channel methods, S. Ritz) addressing nitrate removal in the Elbe.**

On the other hand, if an apparent fractionation factor is calculated for nitrite consumption, why not for nitrate removal as well (which the authors claim is through assimilation only)?

**The reviewer´s suggestion is a good idea to improve the manuscript and have one additional isotope effect. We have calculated the isotope effect of nitrate using an open system approach (see above). The calculated isotope effect for nitrate decrease after discharge peak (7 samples) results in $^{15}\varepsilon$ of -4.0±0.1‰ ($R^2$= 0.90) and $^{18}\varepsilon$ is -5.3±0.1‰, $R^2$= 0.93. This is on the low end of isotope effects reported for nitrate assimilation (Granger et al., 2004; Needoba and Harrison, 2004; Waser et al., 1998),**

**but fractionation can be affected by residence times, such that the isotope effect is lower when residence times are low (Kendall, 1998):**

Also, the 18O to 15N enrichment during nitrate reduction (assimilatory or dissimilatory) should be independent of mixing/dilution and could be more diagnostic of the actual process affecting nitrate dynamics. The authors calculate an 18O to 15N enrichment ratio of 0.82 (page 6, line 32), which is not characteristic for nitrate assimilation.

**This isotope effect reflects the $^{18}$O to $^{15}$N ratio of 0.82. While this ratio is below the 1:1 ratio from batch culture experiments ((Granger et al., 2004) found values between 0.9 and 1.1), it is typical of the Elbe River. (Deutsch et al., 2009) have calculated a comparable $^{18}$O to $^{15}$N enrichment ratio of 0.89, which is attributed to at least 75% nitrate assimilation. Our deviation could be a hint for nitrification in the water column and addition of depleted N. However, nitrite dynamics are rather complex indeed, and isotope changes are more subtle than for nitrite. We would like to point out that we cannot (and do not) exclude a certain role of denitrification – which will in the revised version also be discussed with reference to nitrite.**

**Specific questions and comments**

The title is rather weak. What does "nitrification as a case study" mean? I would change the title to something like "Nitrification and nitrite isotope composition after an exceptional flood in the Elbe River". You could also change "a major European River" to "the Elbe River".

**We agree that the title might not have been appropriate and will modify it to account for the fact that not only nitrite oxidation might have been active.**

Page 1:

Line 8. Why do we care about a nitrate source from NH4+?

**In our manuscript, the focus is set on nitrification as an important process producing nitrate from ammonium. Nitrification is one source of nitrate besides different N sources like urban waste water, surface runoff, and leachate from agriculture soils (Brion et al., 2004; Van Breemen et al., 2002), where nitrification of ammonium occurs. We care about nitrate because generally, ~ 97.0% of total DIN are present as nitrate, it is a monitoring parameter and a link to denitrification, an ultimate removal pathway of Nr that can disrupt the nitrogen cascade of assimilation-remineralization-assimilation.**

From an environmental perspective the source of total fixed N is important to know.

**We disagree. Budget-wise, Nr might be the most relevant parameter, but from an environmental perspective, it does make a difference whether N is present as nitrate, nitrite, or ammonium.**

Line 9. What do you mean with recycling processes? DNRA?

**Has been rephrased to "N-cycling." Potentially important processes are explained in detail in the introduction (page 2, line 4 and following) and like mentioned above like assimilation-remineralization-nitrification-assimiliation....**

Lines 10-12. Do you mean the isotope effects of ammonium oxidation to nitrite and nitrite oxidation to nitrate? Why do we care about isotope effects associated with nitrite reduction/oxidation if no nitrite accumulates? The isotope effect of ammonium oxidation should be easier to predict if no nitrite accumulates, not more difficult.

**Yes, did mean the isotope effects of ammonium oxidation (to nitrite) and nitrite oxidation (to nitrate). Ammonia oxidation is associated with conventional fractionation, whereas nitrite oxidation fractionates inverse and preferentially oxidizes $^{15}$N. In the oceans, nitrite is an intermediate in some key biological processes and can accumulate to high concentration in the primary nitrite maximum (Lomas and Lipschultz, 2006), and in the secondary nitrite maximum in OMZs (e.g. (Lam et al., 2011). Isotope effects (i.e. bulk isotope effects) are investigated in these cases to unravel N-cycling, and isotopes can be indicative of oxidative versus reductive processes. The occurrence of inverse fractionation can indeed complicate the interpretation of bulk isotope effects. We found that the accumulation of nitrite in the Elbe on this rare occasion provided a unique opportunity to study isotope changes in nature.**

**However, we rephrased this sentence to [....] makes it difficult to study the apparent isotope effect of nitrite removal in natural systems.**

Line 16. Delete "the" before concert.

**Done.**

Line 16. Define SPM.

**Has been clarified to "suspended particulate matter (SPM)".**

Line 18. What do you mean by "internal recycling processes"? DNRA?

**Here, we refer to remineralization and nitrification. We rephrased this in the manuscript. As mentioned above, this is addressed in the introduction.**

Line 18. State that xy% of reactive N in the water column was present as nitrate, which leached from the catchment area. This also contradicts the first sentence of the abstract, which says that nitrification is "an important nitrate source" in the water column.

**Was rephrased to "We found that in the water column, ammonium and nitrite built up during the flood event, whereas nitrate was leached from catchment area and appeared to be subject to assimilation".**

**Nitrate is 97.0 – 99.4% of total reactive nitrogen during the flood. However, nitrification IS important; it is the process generating nitrate from ammonium, which is present in fertilizer, making it indeed an important process. In summer nitrification takes place in the water column as well, as rate measurements (not part of this dataset) prove. Actually, the fact that nitrate is so abundant in itself is proof of nitrification – any ammonium that is produced from remineralization is usually oxidized.**

Line 19. The maximum ammonium concentration stated here does not agree with Fig. 2b. After reading all the paper I think there is a mistake in the legend in Figure 2b. The symbols for ammonium and nitrate concentrations are mixed up?

**We apologize for this mistake. Unfortunately, the symbols were mixed up, this has been corrected.**

Line 20. Write here on what basis you conclude that the increase of ammonium and nitrite concentrations is due to remineralization and nitrification.

**This is discussed in detail in the manuscript (amongst others page 7, line 6, 16). Ammonium is tightly bound to clay particles due to its positive charge, such that an external source is unlikely. Nitrite is usually immediately detoxified by oxidation. The succession of ammonium and nitrite is a strong indication for nitrite stemming from ammonium. However, in the abstract, we feel that a detailed explanation is not appropriate – this is done in the manuscript itself.**

Line 25. What is the justification then to use a Rayleigh model?

**We refrained from using this Rayleigh model – see detailed response above.**

Line 26. I think this statement is not justified. The data are not a first approximation of the isotope effect of nitrite oxidation in a natural environment. First, as stated one line above, we have co-occurring nitrite production, and second, we have not seen a proof that nitrite is indeed oxidized (and not assimilated/reduced).

**It has been changed to "Our data show a unique co-occurrence of ammonium, nitrite and nitrate in a river system during an extreme event in summer. Based on this, we calculated an apparent isotope effect of 10.0±0.1‰ during nitrite removal and evaluate different scenarios that may cause this isotope effect."**

Line 33. Write "dissolved inorganic nitrogen (DIN)".

**Done, as suggested.**

Line 34. What about Nr inputs? Ammonium input has been reduced, so now the same amount of Nr is introduced to the system as nitrate?

**No, this is not the case. Nitrate input decreased by 48% (54000 tN a-1 in 1986 to 28000 tN a-1 in 2008) and ammonium input decreased by 93% (12000 tN a-1 in 1986 to 1400 tN a-1 in 2008), resulting in an overall decrease of about 56% (Bergemann and Gaumert, 2008).**

Page 2:

Line 4. Sentence starting with "nevertheless" makes no sense. Nitrate is part of DIN.

**Has been clarified and changed to "Today, the riverine DIN load consists mainly of nitrate, which stems from urban waste water, surface runoff, and leachate from agriculture soils (Brion et al., 2004; Van Breemen et al., 2002). However, nitrate regeneration in rivers can also modify DIN loads (Middelburg and Nieuwenhuize, 2001)…"**

Line 15. The fractionation factor can be calculated differently, not only approximated using a Rayleigh distillation model.

**See discussion above.**

Line 16. (Half-)sentence makes no sense. What are the "individual uptake processes"?

**"Individual uptake processes" refers to assimilation, denitrification etc., i.e. all enzymatically catalyzed nitrogen transformation processes that are relevant in N-cycling.**

Line 26. What do you mean by "relatively healthy"?

**Has been rephrased to "actively nitrifying environments". In "healthy" environments, the co-occurring processes interact and toxic nitrite does not accumulate.**

Page 3:

Line 16. How were samples taken? From the shore? From a boat?

**Has been clarified to "from a quay wall at the shore".**

Line 21. Define SPM.

**Has been clarified to "suspended particulate matter (SPM)".**

Line 22-24. Please give a detection limit and analytic precision for nitrate, nitrite, and ammonium concentration measurements.

**Ammonia analysis has a detection limit of 0.5 µmol L$^{-1}$ and an analytic precision of 0.1 µmol L$^{-1}$. Nitrite analysis has a detection limit of 0.1 µmol L$^{-1}$ and an analytic precision of 0.1 µmol L$^{-1}$. Nitrate analysis has a detection limit of 1.0 µmol L$^{-1}$ and an analytic precision of 0.1 µmol L$^{-1}$.**

Line 26. "Dual nitrate plus nitrite…"

**Done, as suggested.**

Page 4:

Line 1. How was ammonium converted to nitrite? Does this mean that the isotope measurements include the naturally abundant nitrite? Why was the produced nitrite not reduced to N2O with the bacterial method as stated above, but with sodium azide instead?

**The citied reference includes detailed descriptions of the method. It has been clarified to "For analysis of the ammonium isotopic composition, nitrite is first removed by addition of sulfamic acid (Granger and Sigman, 2009). Afterwards, ammonium was chemically converted to nitrite using basic hypobromite oxidation and then reduced to N$_2$O using sodium azide (Zhang et al., 2007)." The denitrifier method is prohibited by using these toxic chemicals.**

Line 9. Delete ", which both by definition have a delta value of 0‰". This is obvious.

**Done, as suggested.**

Line 18. What are the "additional standards"?

**We now specified this in the manuscript (we used commercially available KNO$_3$, KNO$_2$, and NaNO$_2$ salts). δ$^{15}$N-values are determined using our EA-IRMS.**

Lines 19-23. See my comments above. State here under what conditions a Rayleigh model can be used. Why do you think these conditions are met in your system? What could such an apparent fractionation factor tell us, what certainly not. What are the limitations? It is too simplistic (even wrong) to just say that a fractionation can be calculated this way in your system.

**As specified in the general comments, we agree with the concern of using the Rayleigh assumption. This conventional normal fractionation factor indicates a removal of light $^{14}$N from the nitrite pool resulting in an increasing pool signature or this could be additionally coupled with heavy $^{15}$N-nitrite from ammonia oxidation.**

Page 5:

Line 6. I guess the cited 2015 is a mistake, if the here presented data are from 2013?

**We apologize for this typo, corrected in the revised version (2013).**

Line 12. In the abstract it was a maximum of 4.5 μmol L-1. Overall, there is only a 1.1 μmol L-1 decrease over 6 days. What is the precision of your measurements?

**The discrepancy was due to a rounding error, has been changed in the abstract to 4.4 μmol L$^{-1}$. The precision of the nitrite analysis is 0.1 μmol L$^{-1}$. We have clarified this following your suggestions for page 3 lines 22-24.**

Line 15. Logic is not clear. First nitrite is discussed, then ammonium, then nitrite again.

**The succession has been changed for clarity.**

Line 16. What is this limit of 2.7 μmol L-1?

**The number was included merely for precision, this is no specified detection limit is 0.1 μmol L$^{-1}$. It has been clarified in the "Materials and Methods" chapter as your suggestions for page 3 lines 22 – 24. Sentence has been changed (see below).**

Line 16. We see an increase in nitrite concentrations and we also see a decrease in oxygen concentrations. The nitrite concentration increase is not necessarily due to an oxygen concentration decrease.

**The sentence has been changed to "The further elevation of nitrite concentration coincides with a decrease in oxygen (<7.6 mg L$^{-1}$)."**

Line 27. Here and throughout the manuscript: define "flood" and use it consistently. Or what does "before the flood" mean here?

**We defined "flood conditions" on page 4 line 26 as discharge >3000 m$^3$s$^{-1}$. Sections like "before the flood" should make it easier for the reader to follow.**

Line 27. I don't see the nitrite concentration increase in the figure. Can you give numbers?

**Has been clarified to "from 1.6 to 1.8 μmol L$^{-1}$"**

Lines 28-29. The isotope minimum does not correspond to the concentration maximum. Why?

**We assume that the nitrite maximum and isotope minimum do not entirely match because of the influence of the isotope value of the substrate ammonium, which is 2‰ shortly before the lowest nitrite isotope value and increases afterwards.**

Line 30. See comment above and avoid saying "calculated" fractionation factor, as if this would be a real/robust value. With a Rayleigh model you can at best approximate a fractionation between a substrate and a product, and only under certain assumptions (no co-occurring production, product continuously removed from a well-mixed system, etc.)

**We have changed the mode into an open-system approach and rephrased to "apparent isotope effect".**

Line 35. Is the legend in Fig. 2b mixed up? Otherwise this statement makes no sense.

**We apologize for this mistake. Unfortunately, the symbols were mixed up. This has been corrected.**

Line 36. Explain why you compare δ15N-NH4+ to [SPM]. Also, I do see some zic-zac correlation between the two. The second half of the sentence ", but not so much for δ15N-SPM" makes no sense.

**SPM is the source for remineralization to ammonium and therefore ammonium isotope values change with SPM concentration. We will briefly explain this in the revised manuscript version.**

Page 6:

Line 2. What means "initial"? Your first measurement?

**Indeed, we refer to the sample from 11 of June, where ammonium concentration was firstly high enough for isotope analysis.**

Line 8. Legend in Fig. 2b is mixed up. Also give here the actual nitrate concentrations.

**We apologize for the mistake. It has been changed to "With increasing discharge, nitrate concentrations decreased from 270 to 228 µmol L$^{-1}$…"**

Line 10. The sentence makes no sense. Or what does "an increase in SPM peak" mean?

**Has been clarified to "On 9 June, lowest nitrate concentration coincides with a SPM peak of 70 mg L$^{-1}$. After 10 June, nitrate concentrations increased with discharge."**

Line 10. Sentence starting with "both…" needs to be rephrased.

**Has been changed to "Phenomena like this can be attributed…"**

Line 13. How do you explain the time shift between the [SPM] and [NO3-] peak? If the same mechanism is causing the peak (leaking from agricultural soils) I would expect a simultaneous peak?

**This is a characteristic succession for a flood (Baborowski et al., 2004; Pepelnik et al., 2005). SPM is pushed in front of the crest of the flood, whereas nutrients concentrations increase later, because they are dissolved in the water, and also, depending on the substance investigated, because they may be drained from the catchment area.**

Lines 13-14. I don't understand the point made here. Can you elaborate more on why you expect a δ15N-SPM decrease from 7.8 to 6.2‰? And why would along this argumentation δ15N-SPM increase after 13 June and then decrease again after 16 June?

**Isotope values are source specific and change with turnover processes due to fractionation. Fresh, unprocessed material has lower isotope values, because no less fractionation occurred. Furthermore, decreasing δ$^{15}$N-SPM values reflect another source, if suspended particulate material from the catchment enters the river. Terrestrial organic matter has a δ$^{15}$N-value of about 3.5‰, which is significantly lower than riverine SPM with δ$^{15}$N of about 9‰ (Middelburg and Nieuwenhuize, 1998) and references therein). Further, we have analyzed C/N ratios, which have high values of up to 10 with simultaneous SPM concentration peak of 70.5 mg L$^{-1}$ on 9 June. This indicates a decrease of fresh organic material like phytoplankton and an increase of more refracted terrestrial organic matter. Terrestrial organic matter has a δ$^{15}$N-value**

**of about 3.5‰, which is significantly lower than riverine SPM with δ$^{15}$N of about 9‰ ((Middelburg and Nieuwenhuize, 1998) and references therein).**

**The increase and decrease of δ$^{15}$N-SPM can be explained with an increase of C/N ratio to 10 on 16 of June. Afterwards, the C/N ratio again decreases, coupled with a decreasing δ$^{15}$N-SPM. This indicates newly build phytoplankton with lower δ$^{15}$N-values, because they fractionate the nitrate source. Vice versa, nitrate δ$^{15}$N-values increase from about 8.1 to 8.8‰, which also show the increasing importance of nitrate assimilation by phytoplankton.**

Line 21. "reduced biological activity" relative to what? To summer, when we observe average δ15N-NO3- values of … ‰?

**Relative to summer conditions indeed. Has been clarified to "In the Elbe River, δ$^{15}$N-NO$_3^-$ and δ$^{18}$O-NO$_3^-$ values are up to 16.5 and 7.6‰ in June 2006, respectively, and in January 2006 δ$^{15}$N-NO$_3^-$ and δ$^{18}$O-NO$_3^-$ values are 9.3 and 0.2‰, respectively (Johannsen et al., 2008). Another reference sited mean summer δ$^{15}$N-NO$_3^-$ and δ$^{18}$O-NO$_3^-$ values of 18.0±2.7 and 7.6±2.7‰, respectively, and winter δ$^{15}$N-NO$_3^-$ and δ$^{18}$O-NO$_3^-$ values are 9.3±0.7 and 0.8±0.6‰, respectively (June 2005–December 2007, (Schlarbaum et al., 2011). During the flood in June, we see δ$^{15}$N-NO$_3^-$ ranges between 7.4 – 9‰ and δ$^{18}$O-NO$_3^-$ between 2.1 – 3.9‰, which is close to winter values (Fig. 2c)."**

Line 22. What "then"? in summer?

**Has been changed to "In summer and under normal flow conditions, nitrate concentration decreases due to assimilation and biomass production (Fig. 1)."**

Line 23. Add a reference to where it was shown that nitrate concentrations decrease due to assimilation only (and no denitrification).

**Johannsen et al. (2008) showed that assimilation is the main reason for the decrease, as did Deutsch et al. (2009). References will be included.**

Line 29. Rising oxygen concentrations to above saturation? Otherwise it would be more interesting to know what caused the [O2] decrease before the system is ventilated again.

**Oxygen concentration increases from 6.0 mg L$^{-1}$ (corresponding to 63% saturation) to an intermediate maximum of 7.6 mg L$^{-1}$. The dissolved oxygen concentration is clearly correlated to discharge, because initial oxygen concentration is about 10 mg L$^{-1}$, corresponding to an oxygen saturation of about 100% and more, depending on the time of day of sampling. Furthermore, increasing discharge increase turbidity and decrease light irradiation, which weaken phytoplankton activity and their oxygen production. After the 7.6 mg L$^{-1}$ peak, [O$_2$] decreases, accompanied by a strong increase in water temperature (Fig. 2a).**

Line 30. Can you show here this correlation?

**As illustrated in figure 1 of the response letter, [NO$_3^-$] vs. δ$^{15}$N-NO$_3^-$ correlate on a slope of -0.014 with R² of 0.88 and [NO$_3^-$] vs. δ$^{18}$O-NO$_3^-$ with a slope of -0.019 with R$^2$ of 0.92. This figure will not be added to the manuscript.**

[Figure]

Fig. 1 Nitrate concentration versus $\delta^{15}N\text{-}NO_3^-$ and $\delta^{18}O\text{-}NO_3^-$ of water samples from 13 to 20 June 2013.

Line 32. Can you add a plot showing the $\delta18O$- to $\delta15N\text{-}NO3$- increase?

**Figure 2 of the response letter illustrates the last 9 sample values and will be added to the manuscript as your suggestion to a "figure 3" for page 6, line 32.**

[Figure]

Fig. 2 $\delta^{15}N\text{-}NO_3^-$ versus $\delta^{18}O\text{-}NO_3^-$ of water samples from 13 to 27 June 2013.

Line 33. No! Exactly the opposite is the case. 0.82 is significantly different from $1.00 \pm 0.01$, which could have been indicative of assimilation (Granger et al. 2004). The lower value of 0.82 is interesting though. Can you elaborate on the reason for the observed 18O to 15N enrichment ratio?

**We agree 0.82±0.06 is different from the 1:1 enrichment suggested by (Granger et al., 2004) for nitrate assimilation. We see a larger enrichment in the oxygen isotopes than in the nitrogen isotopes. This may be a sign of addition of depleted N from nitrification, or some riparian denitrification.**

**However, we disagree that this rules out that assimilation is the dominant uptake pathway for nitrate (see also response to this issue above).**

**Page 7:**

Line 2. I am not convinced at all. You did not prove that nitrate concentrations decreased due to assimilation only. In contrary, the 18O to 15N enrichment ratio seems to be indicative of other processes.

**The isotope values would not change, if the water parcel with higher nitrate concentrations discharges and water with the same source but lower concentrations follows. Therefore, the isotope signature of the nitrate from the catchment or in the river has to change due to fractionation. The possible process, where dual isotope values increase is assimilation on a slope on 1, and nitrification would lower this ratio towards 0.82 as mentioned above.**

Line 2. What do you mean by "explained by hydrographic properties"?

**Has been clarified to "hydrographic properties associate with increasing discharge"**

Line 3. What is "not quite the case"? Why can nitrite and ammonium concentrations not be explained by assimilation and "hydrographic properties"?

**In our opinion, nitrite and ammonium changes cannot be explained with discharge and assimilation (only), and especially for ammonium, there is no clear link to discharge. In the revised manuscript, different reasons for varying concentrations are given. Ammonium and nitrite do not derive from the catchment, where nitrate, in contrast, always is present, but appear to be built during the flood event (cf. page 6, line 17). We suggested that ammonium stems from mineralization and nitrite stems from nitrification. We will now also address an additional nitrite source; nitrite may be linked to the breakdown of phytoplankton activity, analogous to the primary nitrite maximum in oceans. However, the isotope difference to ammonium is in the range expected for the isotope effect of ammonium oxidation, this suggests that ammonium is a relevant nitrite source. In either case, it stems from a disruption of normal biogeochemical cycles due to the flood. We will clarify this in the manuscript. The authors suggest little nitrite assimilation of phytoplankton, because nitrate is available and preferentially consumed.**

Lines 4-5. There are lots of conclusions in this sentence without any argumentation, underlying data, or proofs. Ammonium and nitrite are also substrates for assimilation and anammox, and nitrite can be reduced to N2O, N2, or ammonium. A decrease in ammonium and nitrite concentrations is hence not a proof for nitrification.

**The main conclusion is that nitrification is responsible. We widened the focus in the revised version to include other sinks like denitrification. However, anammox is likely**

**not important in an organic-rich shallow environment like the Elbe, and we addressed above that DNRA also appears to be of little importance (Burgin and Hamilton 2007, S. Ritz pers. comm.)**

Line 6. Start here with a short discussion of where the NH4+ can come from, instead of just claiming that all of it is originating from SPM remineralization.

**Page 7 line 16 – 18 is moved up, like suggested.**

Line 8. With "initially" you mean your first measurement?

**"Initially" means the first sample on 11 June, where ammonium concentration is high enough for isotope analysis.**

Lines 14-15. Sentence not needed. What do you mean by "this also explains"?

**It will be clarified in the revised version. $\delta^{15}$N-SPM is 6.6‰ and simultaneously $\delta^{15}$N-NH$_4^+$ is 2.0‰. We assume that the difference of 4.6‰ is due to fractionation during remineralization.**

Line 14. "…between δ15N of dissolved…"

**Done, as suggested.**

Lines 16-18. Move this discussion up. (see comment above, line 6). Also, SPM is leaching, so NH4+ could be attached to it and thus leaching as well?

**As mentioned in (Mancino, 1983; Mancino and Troll, 1990) ammonium loss by leachate is probably negligible, because in the catchment it is either consumed or nitrified. We will stress this in a revised version of the manuscript.**

Line 22. Why are O2 concentrations "low"?

**This has been changed to clarify the correlation between [O$_2$] vs. ammonium and nitrite concentrations. Further above, the sentence "The reduced phytoplankton activity is also indicated by decreasing oxygen concentrations." is added to clarify the origin of decreasing oxygen concentrations.**

Lines 22-23. But phytoplankton activity is not 0, right? Based on what are you so convinced that nitrification is the ammonium sink?

**We cannot rule out assimilation of ammonium by phytoplankton, but we have explained why we regard this as unlikely. Also, the initial difference in isotope values of ammonium and nitrite is in the range expected for ammonia oxidation. The succession of peaks also suggests a significant role of nitrification.**

Line 23. Why do you need to say that AOB are active under low [O2]? What do you mean by low [O2]? 6 mg/l is not really low.

**The decreasing oxygen concentration is a hint for reduced phytoplankton activity and we merely wanted to highlight that nitrifiers are active during flood conditions.**

Line 24. Why can nitrification "obviously" not keep up with remineralization? Why is this obvious?

**Because ammonium accumulates, and the remineralization rate thus is higher than the oxidation rate. Rates and their differences will be discussed in a revised version when we assess nitrite isotope changes and calculate possible sinks (the approach we refer to is in the detailed response above addressing general issues).**

Line 27. Define "low oxygen concentration".

**This has been clarified with an absolute value of "below 7.5 mg L$^{-1}$".**

Line 28. Not really. Only the last five data points (out of 21) show a negative correlation.

**Has been clarified to "From 14 June on…"**

Lines29-30. Where do you see the "newly produced nitrate"? If nitrate is produced it contradicts the first sentence of this paragraph, which says that nitrate dynamics are controlled by assimilation and "hydrographic properties". And again, why do you completely rule out any ammonium assimilation? Why would only nitrate by assimilated and not ammonium?

**The amount of nitrate from remineralization and nitrification is of course small in comparison to the nitrate pool, which is above 97.0%…Therefore, nitrate concentrations are controlled by assimilation and hydrology. It is, as we say above, true that we cannot entirely rule out ammonium assimilation. We will focus the manuscript to stress that normal conditions are disrupted – this is obvious – and that we assume nitrification is a sink for ammonium. It may not be the only one, but certainly is active – as we evaluate in the new isotope calculation mentioned at the beginning.**

Line 30. What causes the lowered [O2]?

**Further above, the sentence "The reduced phytoplankton activity is also indicated by decreasing oxygen concentrations." is added to clarify the origin of decreasing oxygen concentrations.**

Line 31. Within 7 days of or after what?

**Has been clarified to "From 9 June to 15 June".**

Lines 31-32. Again, this is an overstatement. You have not proven that all ammonium is oxidized to nitrate and even less so by AOB.

**We agree, and this is rephrased in the manuscript.**

Line 33. Where is that 0.5 μmol L-1 day-1 coming from? I see a sharp decrease in [NH4+] of about 3 μmol L-1 day-1 on 15 June 2013.

**Right, this is a mistake, which has been changed to 3.8 μmol L$^{-1}$ d$^{-1}$: 14 June at about 5pm 3.4 μmol L$^{-1}$ → 15 June at about 2pm 0.0 μmol L$^{-1}$.**

Line 35. Where is that drop in nitrite concentrations after the flood? In Fig. 2b [NO2-] remains at >3 μmol L-1.

**We here show the flood situation only. The nitrite decrease after the flood is not shown, which we clarify in the manuscript (by adding "not shown in plots").**

Page 8:

Line 1. Title is misleading (wrong) and contradicts your conclusion from the last paragraph on page 9.

**Has been clarified to "isotope effect during nitrite removal"**

Lines 3-4. No input of new nitrite? This directly contradicts your conclusions e.g., on page 9, line3.

**In this paragraph, we explain why we have chosen a "Rayleigh" model, which is associated with no input of new nitrite. This assumption is indeed violated, and the manuscript has been changed to an open-system approach - therefore this paragraph been rewritten.**

Line 5. How to you consider the surface water of a river a "closed system"?

**Of course, rivers are not a "closed system", but that was assumed to calculate the fractionation factor and compare this with laboratory results (which use closed systems). The isotope effect was also calculated using an open system approach. Both calculations have conventional normal fractionation factors. We deleted all closed-system assumptions from the manuscript.**

Line 8. What was in steady-state during the flood?

**We refer to the model name "steady-state" (Sigman et al., 2009). In this assumption, new substrate is continuously supplied and partially consumed. This is in contrast to the closed-system "Rayleigh" assumption. Rates are in steady state at each of the linear sections (increase and decrease) of the nitrite concentration plot.**

Line 10-11. Again, I don't believe that all nitrite is oxidized to nitrate in the water column. What about dilution of the signal with other water masses? Nitrite assimilation? Sedimentary denitrification? (not to mention other processes such as anammox, DNRA in the sediments).

**See general discussion at the beginning.**

Lines 15-25. This is pretty obvious and could be deleted.

**We have shortened this paragraph, but still want to discuss "objections".**

Line 27. Lehmann et al. (2004) do not show that denitrification does not take place in the Elbe River, nor that O2 concentrations were >6mg/L.

**(Lehmann et al., 2004) has of course not investigated the Elbe River, but the Santa Monica Bay. This has been rephrased; the reference was in the wrong place. We apologize for this mistake.**

Lines 26-29. Sentence makes no sense and needs to be rephrased.

**This has been rephrased.**

Lines 27-28. Sedimentary denitrification can be the reason for low apparent isotope effects in the water column.

**Indeed. This is what we refer to – not sure what the reviewer's point is here. Sentence has been rephrased for clarity anyway.**

Lines 29-30. 25% of what? If 25% of nitrate is reduced to N2 (or N2O) it is not negligible!

**As stated above, sedimentary denitrification is not associated with a pronounced isotope effect and thus has no or little influence on isotopes and can be neglected with regards to its isotope effect.**

Line 32. I don't agree. (see comments above).

**As stated, we have discussed this more carefully.**

Page 9:

Lines 21-27. Generally, it is very difficult to interpret the isotope signatures presented in this study without any idea about reaction rates. Could you, instead of assuming a Rayleigh model, develop a simple reaction model, which allows you to reproduce all of your measured data (nitrate, nitrite, ammonium, and SPM concentration and isotope data (at least after 14 June)), based on reaction rate constants as tunable parameters? This would give you very important information on what processes could have been active during the flood. I feel this paper could be improved significantly with such a model.

**We have investigated incubation experiments to determine ammonium oxidation and nitrite oxidation rates in 2012 and can estimate rates of up to 14 μmol L$^{-1}$ d$^{-1}$. However, this does not apply accordingly during flood conditions. Furthermore, fractionation factors differ in wide ranges and a model using so many assumptions would misrepresent the processes and their isotope effects. However, we have investigated a back-of-the-envelope calculation (page 9, line 21), which we have extended in the revised version. Therein, we have tested our hypothesis assuming comparable ammonium and nitrite oxidation rates, the isotope value of ammonium increase with an isotope effect for ammonium oxidation of -35‰, which "dilutes" the nitrite pool with isotopically heavy ammonium, an inverse isotope effect of +10‰ during nitrite oxidation. This results in a conventionally negative isotope effect of about -8‰, which is comparable with the calculated isotope effect during the flood. This will be discussed in more detail in the revision.**

Page 10:

Line 8. What do you mean by "not expressed". It is expressed but overprinted by other effects?

**Yes, it is. We have rephrased the sentence.**

Figures:

Figure 1. Caption. Define "flood". I thought you argued that assimilation was the one and only nitrate sink?

**Flood conditions were defined as "discharge >3000 m$^3$ s$^{-1}$" (page 4, line 26).**

Figure 2b. Legend/symbols got mixed up (see comments above). How do you explain the sharp drop in ammonium concentration on 15 June 2013?

**We apologize for the confusion, the symbols has been changed. 3.4 µmol L$^{-1}$ ammonium can be oxidized by ammonia oxidizing bacteria.**

2c. What is the dashed line? Why are there gaps in the δ15N-SPM profile?

**The lines in figure 2c have been changed into solid once to clarify the progress of values. $\delta^{15}$N-SPM could (due to low N content) not be analyzed in all samples.**

Figure 3. I don't think this figure is of much use. Or explain in the caption (and main text) what we are supposed to learn from it.

**The x-axis has been changed in "oxygen saturation". The figure illustrates that oxygen concentration decreases, which indicates reduced phytoplankton activity, and ammonium and nitrite, which are products and substrate of remineralization and nitrification, increase. We now refer to it in more detail in the manuscript.**

Figure 4. Where is Δδ15N-NO2- defined? Write in caption what [NO2-]initial is.

**We have added a definition for $\Delta\delta^{15}$N-NO$_2^-$ and [NO$_2^-$]$_{initial}$.**

**$\Delta\delta^{15}$N-NO$_2^-$ = $\delta^{15}$N-NO$_2^-$ - $\delta^{15}$N-NO$_{2\ initial}^-$, which is the first nitrite isotope value of the calculation.**

**[NO$_2^-$]$_{initial}$ is the first nitrite concentration value of the calculation.**

Generally: check for typos and text formatting throughout the manuscript and specifically on pages 4-7.

**We have checked and changed typos and text formatting throughout the manuscript one more time.**

**References**

Baborowski, M., von Tumpling, W., and Friese, K.: Behaviour of suspended particulate matter (SPM) and selected trace metals during the 2002 summer flood in the River Elbe (Germany) at Magdeburg monitoring station, Hydrology and Earth System Sciences, 8, 135-150, 2004.

Bergemann, M. and Gaumert, T.: Gewässergütebericht der Elbe 2006. ARGE Elbe. 2008.

Brandes, J. A. and Devol, A. H.: Isotopic fractionation of oxygen and nitrogen in coastal marine sediments, Geochimica et Cosmochimica Acta, 61, 1793-1801, 1997.

Brion, N., Baeyens, W., De Galan, S., Elskens, M., and Laane, R. W.: The North Sea: source or sink for nitrogen and phosphorus to the Atlantic Ocean?, Biogeochemistry, 68, 277-296, 2004.

Buchwald, C. and Casciotti, K. L.: Oxygen isotopic fractionation and exchange during bacterial nitrite oxidation, Limnology and Oceanography, 55, 1064-1074, 2010.

Burgin, A. J. and Hamilton, S. K.: Have we overemphasized the role of denitrification in aquatic ecosystems? A review of nitrate removal pathways, Frontiers in Ecology and the Environment, 5, 89-96, 2007.

Casciotti, K. L.: Inverse kinetic isotope fractionation during bacterial nitrite oxidation, Geochimica et Cosmochimica Acta, 73, 2061-2076, 2009.

Casciotti, K. L., Sigman, D. M., and Ward, B. B.: Linking diversity and stable isotope fractionation in ammonia-oxidizing bacteria, Geomicrobiology Journal, 20, 335-353, 2003.

Collos, Y.: Nitrate uptake, nitrite release and uptake, and new production estimates, Marine Ecology Progress Series, 171, 293-301, 1998.

Delwiche, C. C. and Steyn, P. L.: Nitrogen isotope fractionation in soils and microbial reactions, Environmental Science & Technology, 4, 929-935, 1970.

Deutsch, B., Voss, M., and Fischer, H.: Nitrogen transformation processes in the Elbe River: Distinguishing between assimilation and denitrification by means of stable isotope ratios in nitrate, Aquatic Sciences, 71, 228-237, 2009.

Granger, J. and Sigman, D. M.: Removal of nitrite with sulfamic acid for nitrate N and O isotope analysis with the denitrifier method, Rapid Commun Mass Spectrom, 23, 3753-3762, 2009.

Granger, J., Sigman, D. M., Needoba, J. A., and Harrison, P. J.: Coupled nitrogen and oxygen isotope fractionation of nitrate during assimilation by cultures of marine phytoplankton, Limnology and Oceanography, 49, 1763-1773, 2004.

Guerrero, M. A. and Jones, R. D.: Photoinhibition of marine nitrifying bacteria. II. Dark recovery after monochromatic or polychromatic irradiation, Marine ecology progress series. Oldendorf, 141, 193-198, 1996.

Horrigan, S., Carlucci, A., and Williams, P.: Light inhibition of nitrification in sea-surface films [California], Journal of Marine Research, 1981. 1981.

Johannsen, A., Dähnke, K., and Emeis, K.: Isotopic composition of nitrate in five German rivers discharging into the North Sea, Organic Geochemistry, 39, 1678-1689, 2008.

Kendall, C.: Tracing nitrogen sources and cycling in catchments, Isotope tracers in catchment hydrology, 1, 519-576, 1998.

Lam, P., Jensen, M. M., Kock, A., Lettmann, K. A., Plancherel, Y., Lavik, G., Bange, H. W., and Kuypers, M. M.: Origin and fate of the secondary nitrite maximum in the Arabian Sea, Biogeosciences, 8, 375, 2011.

Lehmann, M. F., Sigman, D. M., and Berelson, W. M.: Coupling the 15 N/14 N and 18 O/16 O of nitrate as a constraint on benthic nitrogen cycling, Marine Chemistry, 88, 1-20, 2004.

Lomas, M. W. and Lipschultz, F.: Forming the primary nitrite maximum: Nitrifiers or phytoplankton?, Limnology and Oceanography, 51, 2453-2467, 2006.

Mancino, C. F.: Studies of the Fate of NO3-and NH4+ Nitrogen from Various Fertilizers on Turfgrasses Grown on Three Different Soil Types, 1983.

Mancino, C. F. and Troll, J.: Nitrate and ammonium leaching losses from N fertilizers applied to Penncross' creeping bentgrass, HortScience, 25, 194-196, 1990.

Mariotti, A., Germon, J., Hubert, P., Kaiser, P., Letolle, R., Tardieux, A., and Tardieux, P.: Experimental determination of nitrogen kinetic isotope fractionation: some principles; illustration for the denitrification and nitrification processes, Plant and soil, 62, 413-430, 1981.

Mariotti, A., Landreau, A., and Simon, B.: 15 N isotope biogeochemistry and natural denitrification process in groundwater: Application to the chalk aquifer of northern France, Geochimica et Cosmochimica Acta, 52, 1869-1878, 1988.

Mariotti, A., Leclerc, A., and Germon, J.: Nitrogen isotope fractionation associated with the NO2-→ N2O step of denitrification in soils, Canadian journal of soil science, 62, 227-241, 1982.

Mengis, M., Schif, S., Harris, M., English, M., Aravena, R., Elgood, R., and MacLean, A.: Multiple geochemical and isotopic approaches for assessing ground water NO3− elimination in a riparian zone, Ground water, 37, 448-457, 1999.

Middelburg, J. and Nieuwenhuize, J.: Nitrogen isotope tracing of dissolved inorganic nitrogen behaviour in tidal estuaries, Estuarine, Coastal and Shelf Science, 53, 385-391, 2001.

Middelburg, J. J. and Nieuwenhuize, J.: Carbon and nitrogen stable isotopes in suspended matter and sediments from the Schelde Estuary, Marine Chemistry, 60, 217-225, 1998.

Needoba, J. A. and Harrison, P. J.: Influence of Low Light and a Light: Dark Cycle on NO3-Uptake, Intracellular NO3-, and Nitrogen Isotope Fractionation by Marine Phytoplankton, Journal of Phycology, 40, 505-516, 2004.

Olson, R. J.: Differential photoinhibition of marine nitrifying bacteria: a possible mechanism for the formation of the primary nitrite maximum, J. mar. Res, 39, 227-238, 1981.

Pepelnik, R., Karrasch, B., Niedergesäß, R., Erbsloeh, B., Mehrens, M., Link, U., Herzog, M., and Prange, A.: Influence of the flooding in 2002 on the plankton and the quality of water and sediment of the River Elbe over its longitudinal profile, Acta hydrochimica et hydrobiologica, 33, 430-448, 2005.

Santoro, A. E. and Casciotti, K. L.: Enrichment and characterization of ammonia-oxidizing archaea from the open ocean: phylogeny, physiology and stable isotope fractionation, The ISME journal, 5, 1796-1808, 2011.

Schlarbaum, T., Dähnke, K., and Emeis, K.: Dissolved and particulate reactive nitrogen in the Elbe River/NW Europe: a 2-yr N-isotope study, Biogeosciences, 8, 3519-3530, 2011.

Schroeder, F.: Water quality in the Elbe estuary: Significance of different processes for the oxygen deficit at Hamburg, Environmental Modeling & Assessment, 2, 73-82, 1997.

Sebilo, M., Billen, G., Grably, M., and Mariotti, A.: Isotopic composition of nitrate-nitrogen as a marker of riparian and benthic denitrification at the scale of the whole Seine River system, Biogeochemistry, 63, 35-51, 2003.

Sigman, D., Karsh, K., and Casciotti, K.: Ocean process tracers: nitrogen isotopes in the ocean, Encyclopedia of ocean science, 2nd edn. Elsevier, Amsterdam, 2009. 2009.

Van Breemen, N., Boyer, E., Goodale, C., Jaworski, N., Paustian, K., Seitzinger, S., Lajtha, K., Mayer, B., Van Dam, D., and Howarth, R.: Where did all the nitrogen go? Fate of nitrogen inputs to large watersheds in the northeastern USA, Biogeochemistry, 57, 267-293, 2002.

Wada, E. and Hattori, A.: Nitrogen isotope effects in the assimilation of inorganic nitrogenous compounds by marine diatoms, Geomicrobiology Journal, 1, 85-101, 1978.

Waser, N. A. D., Yin, K. D., Yu, Z. M., Tada, K., Harrison, P. J., Turpin, D. H., and Calvert, S. E.: Nitrogen isotope fractionation during nitrate, ammonium and urea uptake by marine

diatoms and coccolithophores under various conditions of N availability, Mar Ecol Prog Ser, 169, 29-41, 1998.

Yoshida, N.: 15N-depleted N2O as a product of nitrification, Nature, 335, 528-529, 1988.

Zhang, L., Altabet, M. A., Wu, T., and Hadas, O.: Sensitive measurement of NH4+ 15N/14N (δ15NH4+) at natural abundance levels in fresh and saltwaters, Analytical chemistry, 79, 5297-5303, 2007.

---

## Author Comment (AC2) · 27 May 2016

**Comments to „Nitrification and Nitrite Isotope Fractionation as a Case Study in a major European River" by Juliane Jacob et al. (2016)**

Referee comments in Times, **author responses in Arial**

**The authors have provided the information in the revised manuscript as requested by the reviewer. We thank the anonymous reviewers for their suggestions and the valuable concerns. In parts, the reviewers address similar issues, we decided to address these points jointly in more detail.**

**One concern was that no independent rate measurements and clues on what processes are active/negligible were done and that our assumption that nitrite oxidation was the main nitrite sink might not be valid.**

**With regards to this subject, we reconsidered our data, and we agree that our focus might have been too narrow to account for potential nitrite sinks. Accordingly, we have rephrased "nitrite oxidation" to "nitrite removal", and added a section about other potential sinks, like riparian denitrification, nitrite assimilation by phytoplankton, dilution and source-mixing.**

**Of these four potential processes, we assume that nitrite assimilation as a sink is of lesser importance. Even though the possibility of nitrite assimilation by phytoplankton is commonly accepted (Collos, 1998), it is energetically expensive because phytoplankton would have to reduce four electrons for every molecule of nitrite to assimilate nitrite. Furthermore, this reduction of nitrate to nitrite usually happens within the cell in the cytoplasm and the chloroplast, respectively. A direct assimilation of nitrite requires active transport through the chloroplast membrane and needs additional energy (Lomas and Lipschultz, 2006), making this process unfavorable in the presence of ample nitrate.**

**This process would not bias our isotope calculation, because nitrite assimilation in a pure culture was associated to a very small fractionation factor of -0.7 to +1.6‰ (Wada and Hattori, 1978), and thus would only have a minor influence on the isotope signature in the river.**

**Regarding denitrification, our initial assumption was that it would be negligible in the water column, because the oxygen concentration is above 6 mg $L^{-1}$, and that sedimentary denitrification, while potentially quantitatively important, has little to no impact on isotope values of the water column nitrate pool (Brandes and Devol, 1997; Mariotti et al., 1988; Mariotti et al., 1982). However, in the revised version, we consider riparian denitrification as a nitrite sink, which can have a notable apparent isotope effect (Mengis et al., 1999; Sebilo et al., 2003).**

Dilution with water masses containing lower nitrite concentrations is unlikely because of the changing nitrite and nitrate isotope values. Source-mixing has also not been taken into account because nitrite is generally not abundant in the catchment and is immediately removed due to its toxicity (page 7, line 7).

We suggest the different shaped graphs of ammonium and nitrite concentrations and isotopes are not only influenced by hydrology, but more by biology. AOB and NOB have a different behavior/sensitivity to surface irradiance (Horrigan et al., 1981). NOB are more light sensitive (Olson, 1981) and poorly recover from photoinhibition (Guerrero and Jones, 1996). This could be a reason why nitrite can accumulate and the variations in concentrations and isotope values are less pronounced.

We did indeed not present rate measurements, however, we conducted incubation experiments to determine ammonium oxidation and nitrite oxidation rates over an annual cycle in 2012. We find nitrification rates of 1 to 14 $\mu mol\ L^{-1}\ d^{-1}$ in winter and summer, respectively. However, due to time constraints, these measurements were not done during the flood event. In any case, such rate measurements can only serve as a proof that nitrification is active, because our sampling scheme does not really contain a temporal component, and rates cannot be connected to the isotope changes we see.

Another concern was the calculation of the fractionation factor of nitrite removal, which was based on Rayleigh closed-system equations. In the original manuscript, we decided to use this assumption, because ammonium concentrations are below the detection limit and from this perspective nitrite is the substrate being consumed. However, we reconsidered this and agree with the reviewers that this assumption is not valid in our case, as we also discussed in the original submission when we evaluated ammonium production. Consequently, we replaced the Rayleigh calculation with an open-system assumption (Sigman et al., 2009) in the revised manuscript. Using this approach, we calculated an apparent isotope effect of -10.0±0.1‰, which is still conventional.

Both reviewers suggested that the use of a simple box model or simple reaction model should be constructed to assess rates and processes occurring in the river. We took this into account and intensively discussed modeling options with two colleagues experienced in isotope modeling and in nutrient modeling in the Elbe. Our idea was to include isotopes in a biogeochemical model previously published by Friedhelm Schröder, who intensely studied the Elbe River in the 1980s and 1990s (Schroeder, 1997). However, both colleagues agreed that a model is nearly impossible to build based on one sampling station only; because incoming and outgoing concentrations are basically unknown and no mass balance can be set up.

In response to the reviewers' suggestion, we decided to try what we considered the next best option. As nitrite trends during the flood are smooth and isotope changes follow a linear pattern, we assume that the ratio of nitrite processing pathways is constant, even though we cannot quantify rates. The source signal is the isotope value of the maximum nitrite concentration, and then calculated different scenarios with varying rates of nitrite oxidation, ammonium oxidation and denitrification to reproduce our measured data, assuming isotope effects from the literature.

These fractionation factors vary depending on involved microorganisms and environment (Buchwald and Casciotti, 2010; Casciotti, 2009; Casciotti et al., 2003; Delwiche and Steyn, 1970; Mariotti et al., 1981; Santoro and Casciotti, 2011; Yoshida, 1988), but within a range that appeared plausible, we varied these effects and corresponding rates. One plausible scenario is that we see a mixed signal of riparian denitrification and nitrite oxidation, with a constant replenishment of the ammonium pool from suspended matter. We will discuss these calculations in a revised version.

Anonymous reviewer #2:

**The authors gratefully thank the reviewer for the constructive feedback and in-depth evaluation of our manuscript. We appreciate the detailed comments, which are very valuable, and will incorporate the various suggestions.**

**We wrote a general chapter at the beginning of the response letter, because the reviewers´ comments (nitrite oxidation only, other potential nitrite sinks like nitrite assimilation, denitrification and dilution, box model, Rayleigh fractionation…) were partly similar.**

Primary concern is that rivers are inherently dynamic… Impractical for the authors to isolate a single biogeochemical process within this physically complex and hydrologic system...

**The authors agree with this concern and have refined the interpretation of nutrient concentrations and nitrite removal as mentioned in our general comments at the beginning of the response letter. The main change is that we, as this reviewer suggested, expanded our "back-of the-envelope calculation" to assess isotope dynamics, now including other potential sinks and isotope effects. We will discuss this in detail in the manuscript.**

In fact, the authors acknowledge that the increase in the nitrate and SPM concentrations on the rising limb and crest of the flood reflect changes in sources of watershed inputs (e.g., P6 L10-14).

**Yes, but we assume this is mostly important for nitrate, which is known to be leached from the catchment in high amounts (Bergemann and Gaumert, 2008). The very smooth succession of nitrite concentration values, in comparison to a pretty dynamic nitrate plot, suggests that it is supplied, and later removed, at a constant rate, or at least at a constant ratio of rates. This is a strong indication for a biological source.**

However, since many (most?) of the features being observed by the authors' measurements may indeed be related to changes in the hydrologic and geochemical inputs to this volume – it becomes virtually impossible to assign the biogeochemical changes observed to processes occurring within the box.

**We agree. Unfortunately, this makes a box-model approach impossible, but we have extended our back-of-the-envelope calculation as explained in the general comments.**

Can any other conservative tracers of flow ($\delta 18O$ water, bromide, chloride, major ions, etc.) be measured to help constrain water (and N) sources during the flood hydrograph?

**$Na^+$ and $Cl^-$ concentrations were indeed analyzed at the University of Hamburg. At the onset and after flood conditions, they measured concentrations of 2.18 and 2.23 mmol $L^{-1}$, respectively. These values are slightly higher than maximal 2 mmol $L^{-1}$ expected for fresh water (Appelo and Postma, 2005). This is indicative for manure/fertilizer and road salt, which leaches from the catchment. In the course of the flood, the concentrations decrease and loads increase, which suggests water input from the catchment area. These parameters are not in the manuscript, because due to a lack of endmember and a detailed discussion of different sources, analogous to (Mengis et al., 1999) is not possible.**

I wonder whether a simple box model could be constructed…

> **See general discussion above.**

Another thought is that the overall discussion might benefit from a re-focusing…

> **We agree and as mentioned in our general comments, we have changed the discussion from nitrification only towards nitrite removal, which can be a combination of nitrification, nitrite assimilation and denitrification.**

…they could make reasonably well-constrained estimates of the rates of multiple processes…

> **As written in the general comments, we have unpublished nitrification rates for this station from an annual cycle in 2012, which vary between 1 to 14 µmol $L^{-1}d^{-1}$, but due to the very different hydrological situation we carefully extrapolate them to the flood conditions. However, due to time constraints, these measurements were not done during the flood event. In any case, such rate measurements can only serve as a proof that nitrification is active, because our sampling scheme does not really contain a temporal component, and rates cannot be connected to the isotope changes we see.**

A Rayleigh model cannot be justified here.

> **We agree, it has been changed to an open system approach, see general discussion above.**

Can it be demonstrated that the nitrite concentrations are not the product of low levels of NO3- reduction occurring in sediments/hyporheic exchange/groundwater? In fact, the nitrite concentrations vary in a smooth fashion (in contrast with the NH4+ concentrations, for example) – which to me might suggest more of a hydrologic control on their dynamics.

> **As suggested on page 8 line 26, an influence of sedimentary denitrification to nitrite concentrations is unlikely and groundwater residence time is low (~30 years, (Behrendt et al., 2002) and thus direct influence of groundwater inflow can be expected to be negligible (Montenegro et al., 2000). A hyporheic exchange or input from the catchment area is also unlikely in steady-state systems, because the toxic nitrite is usually immediately detoxified. However, due to the intense atmospheric deposition in the catchment area, we truly cannot fully exclude the possibility of nitrite input from hyporheic flow. We included this sink in our calculation of processes and isotope effects (to be included in the revised version, see general comments).**

> **The smooth variation of nitrite concentrations is more likely indicative for biologic controlled processes than for hydrological processes.**

After much confusion - I think that the Figure 2B legend is wrong.

> **Indeed! We apologize for the mistake and the arising confusion, symbols have been changed.**

The decrease in nitrate concentrations on the falling limb of the flood are explained by phytoplankton assimilation – why could this not also possibly explain the concomitant los of nitrite and the positive isotope excursion?

> **To our knowledge, nitrite assimilation of phytoplankton is usually not measured, even if the possibility is commonly accepted (Collos, 1998). The assimilation of**

**nitrite is energetically expensive, because phytoplankton has to reduce four electrons for every molecule of nitrite. Usually, phytoplankton reduces nitrate to nitrite within the cell in the cytoplasm and the chloroplast, respectively. If phytoplankton would reduce nitrite, active transport of nitrite through the chloroplast membrane would need additional energy (Lomas and Lipschultz, 2006). Phytoplankton would need enough energy from light, which is limited during flood conditions. Assimilation of ammonium cannot fully be excluded, but is assumed to be of minor importance.**

**Specific Comments**

P1 L9: 'bulk isotope effect of nitrification' is not clear and should be defined.

**This has been deleted in course of the reworking. "bulk isotope effect" has meant the isotope effect of both nitrification steps (ammonia and nitrite oxidation) together.**

P1 L11: 'divergent' is unclear

**Has been clarified to "negative isotope effect of ammonia and the positive isotope effect of nitrite oxidation"**

P1 L16: In concert with…

**Done, as suggested.**

P1 L19: from the catchment area

**Done, as suggested.**

P1 L22: I'm not convinced that you can conclude the changes in isotopes are the result of nitrite oxidation only. You should state that this is the 'apparent' isotope effect of nitrite consumption (although this may also not be valid as calculated – e.g., violation of Rayleigh model assumptions).

**Done, as suggested, see general comments.**

P1 L30: has increased 20-fold

**Done, as suggested.**

P2 L7: Or, the nitrate can be simply exported from the watershed.

**The nitrate concentrations decrease after the peak of 280 µmol $L^{-1}$ on 13 June. Taking only the decreasing concentration into account, nitrate could be exported, but isotope values change and indicate biological processes. In general, background concentration in winter is about 300 µmol $L^{-1}$ (page 2, line 29). Furthermore, $\delta^{15}N\text{-}NO_3^-$ and $\delta^{18}O\text{-}NO_3^-$ of unprocessed nitrate are about are 7.8 – 9.3 and 0.8 ‰, respectively (page 7, line 15; (Johannsen et al., 2008; Schlarbaum et al., 2011). During the flood and after peak discharge when biological activity re-started, especially, $\delta^{15}N\text{-}NO_3^-$ and $\delta^{18}O\text{-}NO_3^-$ increase (page 6, line 18 – 19, figure 2c), which would not be the case, if water parcels with nitrate just discharge.**

P2: I think the imperative for understanding nitrification in riverine systems should be better justified – perhaps in terms of its frequent coupling to N loss processes (anammox and denitrification) and ecosystem services.

**We agree and will stress this in a revised version. It is coupled to N-loss indeed, and is an oxygen sink as well.**

P2 L13: Not just nitrogen uptake – but any enzymatically catalyzed nitrogen transformation process.

**The reviewer is right and the sentence has been changed to "During enzymatically catalyzed nitrogen transformation processes…"**

P2 L14: This is somewhat of a colloquial expression – and should be restated to reflect that enzymatically catalyzed processes occurring slightly faster for lighter isotopes that heavy isotopes.

**It has been clarified as "lighter isotopes are catalyzed faster than heavier isotopes and this changes the isotope composition of the source and the reaction product (Kendall, 1998; Mariotti et al., 1981)."**

P2 L15: The Rayleigh model explicitly requires the assumption of a unidirectional process and no replenishment of the reactant pool. It's not clear that this can be assumed.

**The authors agree with the reviewers´ criticism. As the definition of a closed Rayleigh distillation model, it is not applicable in the case of the flood. Therefore we have changed this assumption in the whole manuscript towards an open steady-state model (Sigman et al., 2009).**

P2 L18: obstacle to what?

**The first step of nitrification (oxidation of ammonia to nitrite) is associated with wide ranging fractionation factors, which hampers the use of model assumptions, in combination with the inverse fractionation of nitrite (second step of nitrification), especially. This is the reason, why we have not investigated a box-model – one can generate a variety of results, not knowing, which is confidential.**

P2 L21: and the remaining nitrite

**Has been changed.**

P2 L26: what is meant here by the term 'healthy?'

**The authors mean the Elbe River under actively nitrifying conditions, when toxic nitrite is immediately removed by organisms and do not accumulates. The term "healthy" might have been not quite correct for the Elbe anyway, and is not relevant for our manuscript anyway. We changed this to "actively nitrifiying".**

P2 L37: Phytoplankton are light dependent

**Done, as suggested.**

P3 L8: the second largest river discharging

**Done, as suggested.**

P3 L25: Isotope analyses

**Done, as suggested.**

P5 L4: Previous studies have found

**Done, as suggested.**

P5 L8: Either present data as singular or plural – not both. : : : nitrate concentrations were: : : Nitrite concentrations were <1.2: : : and ammonium concentrations were below the detection limit

**Corrected throughout the manuscript, as suggested.**

P5 L28: Not sure I would say that the nitrite concentrations rose 'quickly' – they seem to evolve more gradually in fact.

**True – we changed this.**

P5 L30: For reasons discussed above, I think it should be stressed here that this is an 'apparent' fractionation factor.

**Has been changed.**

P6 L2: Remove "it is interesting, however" – opinions don't generally belong in a results section.

**Done, as suggested.**

P6 L 20: Although as noted later – the relatively large nitrate pool is far more resistant to isotopic perturbations by biogeochemical processes.

**The authors agree and have clarified this.**

P6 L25: While this may be true – the watershed flooding potentially may have also introduced a nitrate source having a slightly different isotopic composition.

**Has been changed to clarify different flow regimes: "In our study, before the SPM peak, $\delta^{15}N$-$NO_3^-$ and $\delta^{18}O$-$NO_3^-$ are not correlated with $[NO_3^-]$, probably because of water with slightly different isotope signatures from the watershed flooding. Only after the SPM peak $\delta^{15}N$-$NO_3^-$ and $\delta^{18}O$-$NO_3^-$ are negatively correlated with $[NO_3^-]$ ($R^2$ of 0.897 and 0.816, respectively), which pinpoints…".**

**The main source of nitrate under normal flow is agriculture in the watershed (cf. Johannsen et al. 2008), so a new, relevant source appears unlikely.**

P6 L29: As the phytoplankton are recovering – couldn't they be assimilating nitrite?

**As explained in our general comments, phytoplankton theoretically can assimilate nitrite. To our knowledge, this has not been observed in situ and would be energetically much more inefficient then assimilating nitrate, especially, when the surface irradiance has decreased due to high turbidity and energy is limited. Even if it occurred, its isotope effect would be negligible (Wada and Hattori 1978).**

P6 L32: This _15N vs _18O slope is actually much lower than that observed by Granger and colleagues. Could this be indicative of nitrification?

**Yes, this can indeed be a hint for nitrification and has been clarified in the manuscript.**

P7 L3: Why can't phytoplankton be playing role in assimilation of nitrite and/or ammonium?

**We address this in the general comments and our answer above (to P6 L29). We have added a statement in our manuscript.**

P7 L23: But phytoplankton activity was specifically invoked as explaining an increase in N and O isotopes of nitrate and contributing to a drawdown of _100uM nitrate. Thus, it seems hard to discount phytoplankton activity for drawdown of _1uM nitrite and _3uM NH4+.

**We agree that the focus on nitrite oxidation might have been to narrow. We will discuss the role of nitrite assimilation in the revised manuscript, but still assume it to be less important than other sinks like oxidation or denitrification, for the reasons outlined above, in response to reviewers 1 and 2.**

P8 L10: : : : suggesting conventional normal (as opposed to inverse) fractionation during: : :

**Has been changed as suggested.**

P8 L29-30: The contribution of 25% sedimentary denitrification actually seems substantial – and therefore hard to rule out. Also – can the same conclusions of Deutsh et al., 2009 be drawn for the extreme flood conditions of this study? Couldn't flooding act to increase hyporheic exchange?

**Sedimentary denitrification has little to no impact on isotope values (Mariotti et al., 1988; Mariotti et al., 1982). Regarding hyporheic exchange, a model study showed that it can indeed increase during the flood, but this would also lead to a reduced residence time (Boano et al., 2007), which in turn may decrease denitrification rates. However, this interaction is quite speculative, and we will thus not include it in the manuscript.**

P8 L35: As conceptualized by the authors, since the $\delta^{15}N$ of the NH4+ is ~+2permil to begin with (~4permil lower than the SPM $\delta^{15}N$) – the $\delta^{15}N$ of the NO2- produced (in a closed system) would follow the accumulated product equation – and under conditions where NH4+ was being completely oxidized to NO2- - the newly produced NO2- would have a $\delta^{15}N$ of +2permil. In developing the argument about the contribution of heavy nitrite from ammonia oxidation, the authors should be careful to explain how this new nitrite composition will evolve in step with the degree of NH4+ consumption. Initially the new nitrite will have a $\delta^{15}N$ even lower than the existing nitrite, while as NH4+ is consumed – the $\delta^{15}N$ of the newly produced nitrite will approach the original $\delta^{15}N$ of the NH4+ (~+2‰. This value of +2‰ is actually not the 'isotopically enriched' nitrite that seems to be invoked here by the authors. Later on P9 L18, the authors explain how the complete consumption of NH4+ would quantitatively transfer the $\delta^{15}N$ value of the NH4+ pool into the nitrite pool – yet it is not clear whether the authors are using the evolving NH4+ pool as a closed system – or simply invoking the instantaneous product equation at each step. Notably – these values and mass balance estimates will play importantly into their 'back-of-the-envelope' calculations.

**The reviewer here suggests to carefully check the effects of partial or complete consumption of substrates and their effect on the product isotope pool. We agree that this needed to be developed further in the manuscript. Accordingly, we expanded our 'back-of-the-envelope' calculation (and explanations thereof) to account for this. We aimed to develop scenarios that may explain the nitrite isotope and concentration trends we see and will discuss this in detail in a revised version, see also general comments above.**

P9 L21: I think the discussion could be clarified if these calculations were explained in more detail.

**As we outline in our detailed response at the beginning, we expanded this calculation and the interpretation of the results.**

P9 L36: I don't think there is any sort of cryptic ammonium cycle occurring here. More likely, it seems that the authors are just witnessing more 'conventional' N cycling processes (e.g., remineralization, nitrification, assimilation, etc.) from the perspective of nitrite isotopes for the first time in a river.

**Even if ammonium is not abundant, we assumed a very rapid remineralization of ammonium and its immediate consumption (nitrification – assimilation). We rephrased this, basically not referring to a cryptic cycle. However, we assume that the N-cycle is in so far not conventional, as normal links between processes are disrupted, resulting in accumulation of ammonium and nitrite. Of course, the reviewer is right that these processes occur (likely at different rates) in a river also under normal conditions. We now highlight that the flood disrupted normal N-turnover.**

Figure 2b: Caption is wrong?

**It indeed was - we apologize for this mistake, it has been corrected.**

Figure 4b: this should be labeled as 'apparent isotope effect for nitrite consumption' (not nitrite oxidation). As articulated by the authors in the discussion, I don't think you can tie these isotope changes to a single process.

**The authors agree with the reviewer and have changed the to an open system steady-state assumption with a fractionation factor $^{15}\varepsilon$ of -10.0±0.1‰. We also now do not exclusively refer to nitrite oxidation.**

**References**

Appelo, C. A. J. and Postma, D.: Geochemistry, groundwater and pollution, CRC press, 2005.

Behrendt, H., Kornmilch, M., Opitz, D., Schmoll, O., and Scholz, G.: Estimation of the nutrient inputs into river systems–experiences from German rivers, Regional Environmental Change, 3, 107-117, 2002.

Bergemann, M. and Gaumert, T.: Gewässergütebericht der Elbe 2006. ARGE Elbe. 2008.

Boano, F., Revelli, R., and Ridolfi, L.: Bedform-induced hyporheic exchange with unsteady flows, Advances in water resources, 30, 148-156, 2007.

Brandes, J. A. and Devol, A. H.: Isotopic fractionation of oxygen and nitrogen in coastal marine sediments, Geochimica et Cosmochimica Acta, 61, 1793-1801, 1997.

Buchwald, C. and Casciotti, K. L.: Oxygen isotopic fractionation and exchange during bacterial nitrite oxidation, Limnology and Oceanography, 55, 1064-1074, 2010.

Casciotti, K. L.: Inverse kinetic isotope fractionation during bacterial nitrite oxidation, Geochimica et Cosmochimica Acta, 73, 2061-2076, 2009.

Casciotti, K. L., Sigman, D. M., and Ward, B. B.: Linking diversity and stable isotope fractionation in ammonia-oxidizing bacteria, Geomicrobiology Journal, 20, 335-353, 2003.

Collos, Y.: Nitrate uptake, nitrite release and uptake, and new production estimates, Marine Ecology Progress Series, 171, 293-301, 1998.

Delwiche, C. C. and Steyn, P. L.: Nitrogen isotope fractionation in soils and microbial reactions, Environmental Science & Technology, 4, 929-935, 1970.

Guerrero, M. A. and Jones, R. D.: Photoinhibition of marine nitrifying bacteria. II. Dark recovery after monochromatic or polychromatic irradiation, Marine ecology progress series. Oldendorf, 141, 193-198, 1996.

Horrigan, S., Carlucci, A., and Williams, P.: Light inhibition of nitrification in sea-surface films [California], Journal of Marine Research, 1981. 1981.

Johannsen, A., Dähnke, K., and Emeis, K.: Isotopic composition of nitrate in five German rivers discharging into the North Sea, Organic Geochemistry, 39, 1678-1689, 2008.

Kendall, C.: Tracing nitrogen sources and cycling in catchments, Isotope tracers in catchment hydrology, 1, 519-576, 1998.

Lomas, M. W. and Lipschultz, F.: Forming the primary nitrite maximum: Nitrifiers or phytoplankton?, Limnology and Oceanography, 51, 2453-2467, 2006.

Mariotti, A., Germon, J., Hubert, P., Kaiser, P., Letolle, R., Tardieux, A., and Tardieux, P.: Experimental determination of nitrogen kinetic isotope fractionation: some principles; illustration for the denitrification and nitrification processes, Plant and soil, 62, 413-430, 1981.

Mariotti, A., Landreau, A., and Simon, B.: 15 N isotope biogeochemistry and natural denitrification process in groundwater: Application to the chalk aquifer of northern France, Geochimica et Cosmochimica Acta, 52, 1869-1878, 1988.

Mariotti, A., Leclerc, A., and Germon, J.: Nitrogen isotope fractionation associated with the $NO_2-\rightarrow N_2O$ step of denitrification in soils, Canadian journal of soil science, 62, 227-241, 1982.

Mengis, M., Schif, S., Harris, M., English, M., Aravena, R., Elgood, R., and MacLean, A.: Multiple geochemical and isotopic approaches for assessing ground water $NO_3^-$ elimination in a riparian zone, Ground water, 37, 448-457, 1999.

Montenegro, H., Holfelder, T., and Wawra, B.: Modellierung der Austauschprozesse zwischen Oberflächen- und Grundwasser in Flußauen. In: Stoffhaushalt von Auenökosystemen, Springer, 2000.

Olson, R. J.: Differential photoinhibition of marine nitrifying bacteria: a possible mechanism for the formation of the primary nitrite maximum, J. mar. Res, 39, 227-238, 1981.

Santoro, A. E. and Casciotti, K. L.: Enrichment and characterization of ammonia-oxidizing archaea from the open ocean: phylogeny, physiology and stable isotope fractionation, The ISME journal, 5, 1796-1808, 2011.

Schlarbaum, T., Dähnke, K., and Emeis, K.: Dissolved and particulate reactive nitrogen in the Elbe River/NW Europe: a 2-yr N-isotope study, Biogeosciences, 8, 3519-3530, 2011.

Schroeder, F.: Water quality in the Elbe estuary: Significance of different processes for the oxygen deficit at Hamburg, Environmental Modeling & Assessment, 2, 73-82, 1997.

Sebilo, M., Billen, G., Grably, M., and Mariotti, A.: Isotopic composition of nitrate-nitrogen as a marker of riparian and benthic denitrification at the scale of the whole Seine River system, Biogeochemistry, 63, 35-51, 2003.

Sigman, D., Karsh, K., and Casciotti, K.: Ocean process tracers: nitrogen isotopes in the ocean, Encyclopedia of ocean science, 2nd edn. Elsevier, Amsterdam, 2009. 2009.

Wada, E. and Hattori, A.: Nitrogen isotope effects in the assimilation of inorganic nitrogenous compounds by marine diatoms, Geomicrobiology Journal, 1, 85-101, 1978.

Yoshida, N.: 15N-depleted N2O as a product of nitrification, Nature, 335, 528-529, 1988.

---

## Author Comment (AC3) · 27 May 2016

**Comments to „Nitrification and Nitrite Isotope Fractionation as a Case Study in a major European River" by Juliane Jacob et al. (2016)**

Referee comments in Times, **author responses in Arial**

**The authors have provided the information in the revised manuscript as requested by the reviewer. We thank the anonymous reviewers for their suggestions and the valuable concerns. In parts, the reviewers address similar issues, we decided to address these points jointly in more detail.**

**One concern was that no independent rate measurements and clues on what processes are active/negligible were done and that our assumption that nitrite oxidation was the main nitrite sink might not be valid.**

**With regards to this subject, we reconsidered our data, and we agree that our focus might have been too narrow to account for potential nitrite sinks. Accordingly, we have rephrased "nitrite oxidation" to "nitrite removal", and added a section about other potential sinks, like riparian denitrification, nitrite assimilation by phytoplankton, dilution and source-mixing.**

**Of these four potential processes, we assume that nitrite assimilation as a sink is of lesser importance. Even though the possibility of nitrite assimilation by phytoplankton is commonly accepted (Collos, 1998), it is energetically expensive because phytoplankton would have to reduce four electrons for every molecule of nitrite to assimilate nitrite. Furthermore, this reduction of nitrate to nitrite usually happens within the cell in the cytoplasm and the chloroplast, respectively. A direct assimilation of nitrite requires active transport through the chloroplast membrane and needs additional energy (Lomas and Lipschultz, 2006), making this process unfavorable in the presence of ample nitrate.**

**This process would not bias our isotope calculation, because nitrite assimilation in a pure culture was associated to a very small fractionation factor of -0.7 to +1.6‰ (Wada and Hattori, 1978), and thus would only have a minor influence on the isotope signature in the river.**

**Regarding denitrification, our initial assumption was that it would be negligible in the water column, because the oxygen concentration is above 6 mg $L^{-1}$, and that sedimentary denitrification, while potentially quantitatively important, has little to no impact on isotope values of the water column nitrate pool (Brandes and Devol, 1997; Mariotti et al., 1988; Mariotti et al., 1982). However, in the revised version, we consider riparian denitrification as a nitrite sink, which can have a notable apparent isotope effect (Mengis et al., 1999; Sebilo et al., 2003).**

Dilution with water masses containing lower nitrite concentrations is unlikely because of the changing nitrite and nitrate isotope values. Source-mixing has also not been taken into account because nitrite is generally not abundant in the catchment and is immediately removed due to its toxicity (page 7, line 7).

We suggest the different shaped graphs of ammonium and nitrite concentrations and isotopes are not only influenced by hydrology, but more by biology. AOB and NOB have a different behavior/sensitivity to surface irradiance (Horrigan et al., 1981). NOB are more light sensitive (Olson, 1981) and poorly recover from photoinhibition (Guerrero and Jones, 1996). This could be a reason why nitrite can accumulate and the variations in concentrations and isotope values are less pronounced.

We did indeed not present rate measurements, however, we conducted incubation experiments to determine ammonium oxidation and nitrite oxidation rates over an annual cycle in 2012. We find nitrification rates of 1 to 14 $\mu mol\ L^{-1}\ d^{-1}$ in winter and summer, respectively. However, due to time constraints, these measurements were not done during the flood event. In any case, such rate measurements can only serve as a proof that nitrification is active, because our sampling scheme does not really contain a temporal component, and rates cannot be connected to the isotope changes we see.

Another concern was the calculation of the fractionation factor of nitrite removal, which was based on Rayleigh closed-system equations. In the original manuscript, we decided to use this assumption, because ammonium concentrations are below the detection limit and from this perspective nitrite is the substrate being consumed. However, we reconsidered this and agree with the reviewers that this assumption is not valid in our case, as we also discussed in the original submission when we evaluated ammonium production. Consequently, we replaced the Rayleigh calculation with an open-system assumption (Sigman et al., 2009) in the revised manuscript. Using this approach, we calculated an apparent isotope effect of -10.0±0.1‰, which is still conventional.

Both reviewers suggested that the use of a simple box model or simple reaction model should be constructed to assess rates and processes occurring in the river. We took this into account and intensively discussed modeling options with two colleagues experienced in isotope modeling and in nutrient modeling in the Elbe. Our idea was to include isotopes in a biogeochemical model previously published by Friedhelm Schröder, who intensely studied the Elbe River in the 1980s and 1990s (Schroeder, 1997). However, both colleagues agreed that a model is nearly impossible to build based on one sampling station only; because incoming and outgoing concentrations are basically unknown and no mass balance can be set up.

In response to the reviewers' suggestion, we decided to try what we considered the next best option. As nitrite trends during the flood are smooth and isotope changes follow a linear pattern, we assume that the ratio of nitrite processing pathways is constant, even though we cannot quantify rates. The source signal is the isotope value of the maximum nitrite concentration, and then calculated different scenarios with varying rates of nitrite oxidation, ammonium oxidation and denitrification to reproduce our measured data, assuming isotope effects from the literature.

These fractionation factors vary depending on involved microorganisms and environment (Buchwald and Casciotti, 2010; Casciotti, 2009; Casciotti et al., 2003; Delwiche and Steyn, 1970; Mariotti et al., 1981; Santoro and Casciotti, 2011; Yoshida, 1988), but within a range that appeared plausible, we varied these effects and corresponding rates. One plausible scenario is that we see a mixed signal of riparian denitrification and nitrite oxidation, with a constant replenishment of the ammonium pool from suspended matter. We will discuss these calculations in a revised version.

Anonymous Referee #3

**We would like to thank reviewer for the evaluation of our manuscript, and we will implement the suggestions.**

**In the beginning of the response letter, the authors have answered general concerns of all reviewers (nitrite oxidation only, other potential nitrite sinks like nitrite assimilation, denitrification and dilution, box model, Rayleigh fractionation).**

Could the ammonium and nitrite not be imported from the catchment, from internal cycling therein (in soil)? Which aspects of the isotope data enable partitioning of processes that happened in situ vs. the catchment? Does it even matter?

**As mentioned in the manuscript (page 7, line 16 – 18 and references therein), an ammonium and nitrite input from the catchment is unlikely, because of the positive charge of ammonium molecules in combination with adhesion to clay. Nitrite is toxic and not abundant in steady-state systems, because it is rapidly nitrified (page 7, line 7 and references therein). To our knowledge, these mechanisms are active during the flood in the catchment area and ammonium and nitrite in the Elbe River derive from internal processes (remineralization and nitrification). Theoretically, phytoplankton can release nitrite in stress situation (Lomas and Lipschultz, 2006), but an amount cannot be estimated. Therefore, isotope changes of ammonium and nitrite happens in situ. However, ammonium derives from remineralization of suspended particulate matter and the origin cannot be specified. The $\delta^{15}$N-SPM values vary between 8.1 and 6.2‰ (page 6, line 1) and this show a much smaller variability then $\delta^{15}$N-NH$_4^+$ values. Furthermore, the authors would like to point out that the origin of substrate (ammonium, nitrite) is of minor importance for the isotope effect coupled to nitrite removal. In any case, though, we now included other source and sink processes, but point out that nitrification is an important ammonium sink (and thus nitrite source). This is supported by rate measurements done in the river, where we find nitrification rates of up to 14 µmol L$^{-1}$ d$^{-1}$. We mention these (yet unpublished) data in the manuscript now, but would like to point out here that they can unfortunately not be directly linked to the isotope values we measure at this site, because the measurement at one site has no temporal component to it.**

Could the authors not generate plausible scenarios of nitrite production/oxidation and associated isotope effects that could constrain the relative fluxes, given the measured isotope composition of ammonium and nitrite? I realize the range of solutions may be too broad, but perhaps some scenarios could be ruled out with such an exercise.

**The authors have extended their back-of-the-envelope calculation (page 9, line 21) as mentioned in the general comments.**

The isotope composition of nitrite in the environment is implicitly the result of multiple co-incident reactions, each of which is associated with an isotope effect. It's self-evident that a single Rayleigh fit to NO2 consumption will not describe a single uni-directional reaction on said NO2, which does not mean that culture results cannot be extrapolated to the environment. What an odd conclusion! I urge the authors to refine this conclusion so as to appear less incongruous.

**We agree with the concern of using a Rayleigh model and have changed this as mentioned in the general comments.**

In large parts, nitrate derives from the catchment area (page 6, line 10 – 13) and isotope changes are within a narrow range because of reduced phytoplankton assimilation (page 6, line 26).

However, after revision of the manuscript, the authors would attenuate the statement of "no inverse isotope fractionation during nitrite oxidation" to something along the line of "conventional isotope fractionation during nitrite concentration removal".

**References**

Brandes, J. A. and Devol, A. H.: Isotopic fractionation of oxygen and nitrogen in coastal marine sediments, Geochimica et Cosmochimica Acta, 61, 1793-1801, 1997.

Buchwald, C. and Casciotti, K. L.: Oxygen isotopic fractionation and exchange during bacterial nitrite oxidation, Limnology and Oceanography, 55, 1064-1074, 2010.

Casciotti, K. L.: Inverse kinetic isotope fractionation during bacterial nitrite oxidation, Geochimica et Cosmochimica Acta, 73, 2061-2076, 2009.

Casciotti, K. L., Sigman, D. M., and Ward, B. B.: Linking diversity and stable isotope fractionation in ammonia-oxidizing bacteria, Geomicrobiology Journal, 20, 335-353, 2003.

Collos, Y.: Nitrate uptake, nitrite release and uptake, and new production estimates, Marine Ecology Progress Series, 171, 293-301, 1998.

Delwiche, C. C. and Steyn, P. L.: Nitrogen isotope fractionation in soils and microbial reactions, Environmental Science & Technology, 4, 929-935, 1970.

Guerrero, M. A. and Jones, R. D.: Photoinhibition of marine nitrifying bacteria. II. Dark recovery after monochromatic or polychromatic irradiation, Marine ecology progress series. Oldendorf, 141, 193-198, 1996.

Horrigan, S., Carlucci, A., and Williams, P.: Light inhibition of nitrification in sea-surface films [California], Journal of Marine Research, 1981. 1981.

Lomas, M. W. and Lipschultz, F.: Forming the primary nitrite maximum: Nitrifiers or phytoplankton?, Limnology and Oceanography, 51, 2453-2467, 2006.

Mariotti, A., Germon, J., Hubert, P., Kaiser, P., Letolle, R., Tardieux, A., and Tardieux, P.: Experimental determination of nitrogen kinetic isotope fractionation: some principles; illustration for the denitrification and nitrification processes, Plant and soil, 62, 413-430, 1981.

Mariotti, A., Landreau, A., and Simon, B.: 15 N isotope biogeochemistry and natural denitrification process in groundwater: Application to the chalk aquifer of northern France, Geochimica et Cosmochimica Acta, 52, 1869-1878, 1988.

Mariotti, A., Leclerc, A., and Germon, J.: Nitrogen isotope fractionation associated with the NO2-→ N2O step of denitrification in soils, Canadian journal of soil science, 62, 227-241, 1982.

Mengis, M., Schif, S., Harris, M., English, M., Aravena, R., Elgood, R., and MacLean, A.: Multiple geochemical and isotopic approaches for assessing ground water NO3− elimination in a riparian zone, Ground water, 37, 448-457, 1999.

Olson, R. J.: Differential photoinhibition of marine nitrifying bacteria: a possible mechanism for the formation of the primary nitrite maximum, J. mar. Res, 39, 227-238, 1981.

Santoro, A. E. and Casciotti, K. L.: Enrichment and characterization of ammonia-oxidizing archaea from the open ocean: phylogeny, physiology and stable isotope fractionation, The ISME journal, 5, 1796-1808, 2011.

Schroeder, F.: Water quality in the Elbe estuary: Significance of different processes for the oxygen deficit at Hamburg, Environmental Modeling & Assessment, 2, 73-82, 1997.

Sebilo, M., Billen, G., Grably, M., and Mariotti, A.: Isotopic composition of nitrate-nitrogen as a marker of riparian and benthic denitrification at the scale of the whole Seine River system, Biogeochemistry, 63, 35-51, 2003.

Sigman, D., Karsh, K., and Casciotti, K.: Ocean process tracers: nitrogen isotopes in the ocean, Encyclopedia of ocean science, 2nd edn. Elsevier, Amsterdam, 2009. 2009.

Wada, E. and Hattori, A.: Nitrogen isotope effects in the assimilation of inorganic nitrogenous compounds by marine diatoms, Geomicrobiology Journal, 1, 85-101, 1978.

Yoshida, N.: 15N-depleted N2O as a product of nitrification, Nature, 335, 528-529, 1988.

---

## Author Response (AR1)

**Point-by-Point-Reply to „Nitrification and Nitrite Isotope Fractionation as a Case Study in a major European River" by Juliane Jacob et al. (2016)**

All three reviewers shared main concerns with regards to our manuscript, which is that we have only focused on nitrification, that we should use an open-system instead of a Rayleigh approach to calculate isotope effects and that we should use a simple box-model to evaluate nitrite sinks. We addressed these issues thoroughly, to the extent that large sections of the discussion have been changed. In detail, we included the following changes:

- Most importantly, we re-focused the overall discussion from assuming nitrite oxidation as a sole nitrite sink to an evaluation of nitrite consumption as a whole. We agree that our focus might have been too narrow to account for potential nitrite sinks. Accordingly, we have rephrased "nitrite oxidation" to "nitrite consumption", and added a section about other potential sinks, like riparian denitrification, nitrite assimilation by phytoplankton, dilution and source-mixing (e.g. page 8, line 27; page 9, line 6). Based on isotope dynamics and due to limited evidence for nitrite assimilation by phytoplankton, we conclude, that of these four potential processes, nitrite assimilation is a sink of lesser importance (e.g. page 8, line 23). In contrast, riparian denitrification is a potential nitrite sink, that can have a notable apparent isotope effect (Mengis et al., 1999; Sebilo et al., 2003), page 9, line 17), and we included its role in the discussion (e.g. chapter 4.3).

- We address mixing and dilution effects now in more detail in the manuscript, mainly based on SPM and nitrate dynamics (chapter 4.1). Regarding nitrite, though, we regard dilution and source mixing as unlikely, because nitrite is generally not abundant in the catchment and is immediately removed due to its toxicity (see also original response letter, and manuscript page 8, line 17).

- As suggested by reviewer #1 and #2, we investigated a box-model. In three scenarios, we evaluate the sinks based on isotope effects of coupled riparian denitrification, nitrite oxidation and ammonium assimilation (chapter 4.3):
    - Scenario 1 takes only nitrite consumption due to nitrification and riparian denitrification with divergent isotope effects into account, which results in 22% nitrification. This is somewhat unlikely, because ammonium remineralization and ammonium oxidation thus would not occur (page 9, line 24 et. seq.).
    - Scenario 2 considers constant supply of ammonium with an isotope value of about 4.5‰ from remineralization of SPM and nitrite formation from ammonium. This results in 31% nitrification and 69% denitrification (page 9, line 33 et. seq.).
    - Scenario 3 takes the theory of "cryptic" ammonium cycling" into account. Ammonium is consumed during phytoplankton assimilation and successively gets

enriched in $^{15}$N. As a result, the contribution of nitrite oxidation increases to 36% versus 64% denitrification (page 10, line 14 et. seq.).

- Another concern was the calculation of the fractionation factor of nitrite consumption. We replaced the Rayleigh calculation with an open-system assumption (Sigman et al., 2009). The apparent isotope effect during nitrite consumption is -10.0±0.1‰ (page 6, line 8). Furthermore, isotope effects during nitrate consumption after phytoplankton recovery were calculated, with $^{15}\varepsilon$ of -4.0±0.1 and R² of 0.89, as well as $^{18}\varepsilon$ -5.3±0.1 and R² of 0.92 (page 7, line 18 – 19) and discussed in chapter 4.1.
- We suggest ammonium and nitrite concentrations and isotopes are mainly influenced by biology and minor by hydrology. AOB and NOB have a different behavior/sensitivity to surface irradiance (Horrigan et al., 1981). NOB are more light sensitive (Olson, 1981) and poorly recover from photoinhibition (Guerrero and Jones, 1996). This could be a reason why nitrite can accumulate and the variations in concentrations and isotope values are less pronounced.
- Reviewer #2 asked for nitrification rates. We did indeed not present rate measurements, however, we conducted incubation experiments to determine ammonium oxidation and nitrite oxidation rates over an annual cycle in 2012, which we serve as a proof for nitrification.
- Another point was the ratio of $\delta^{15}$N to $\delta^{18}$O of nitrate, which is 0.82 and deviates from the 1:1 slope associated with assimilation (Granger et al., 2004). (Deutsch et al., 2009) have calculated a comparable enrichment ratio of 0.89, which is attributed to at least 75% nitrate assimilation. Our deviation could be a hint for nitrification in the water column and addition of depleted N. However, nitrite dynamics are rather complex indeed, and isotope changes are more subtle than for nitrite.
- The term "calculated fractionation factor" has been changed to "apparent isotope effect".
- The title has been changed into "Isotope Effects of coupled Nitrification and Denitrification during a River Flood Event"
- Figure 1 is deleted
- Figure 2a – c → Figure 1 a – c has a different color and legend
- Figure 3 → Figure 2 has a changed x-axis (now oxygen saturation [%])
- New figure 3 shows $\delta^{15}$N vs. $\delta^{18}$O of nitrate with a slope of 0.82 and R² of 0.96.
- Figure 4 has a changed x-axis (now (1 - $f$) following an open-system approach) with a slope of -10.0±0.1‰ and R² of 0.97.
- New figure 5 shows $\delta^{15}$N and $\delta^{18}$O of nitrate versus (1 – $f$) and indicates the isotope effects during nitrate consumption.

All specific comments are approved and we refer to the original response letter including very detailed comments.

[revised manuscript text omitted]

**8 Figures**

Summer conditions at normal discharge

[Figure]

Hypothesized flood conditions

[Figure]

Fig. 1 →

[Figure]

**Figure    1a,        b,        c1**

[Figure]

Figure 2a, 2b, 2c

[Figure]

**New figure 2**

[Figure]

→   New figure 3

[Figure]

→ deleted!

[Figure]

5 **New f 3**

[Figure]

➔ **New f 4**

**9 Figure Captions**

Figure 1a

 Discharge, dissolved oxygen concentration, and SPM concentration of the Elbe River water samples  from 6 to 20 June 2013. Flood conditions occur with discharge >3000 $m^3 s^{-1}$.

Figure 1b Ammonium, nitrite, and nitrate concentrations in the Elbe River in the course of the flood. Calculation of the fractionation factor is based on filled  data points.

Figure 1c Ammonium, nitrite, nitrate, and SPM isotope values in the course of the flood. Calculation of the fractionation factor is based on filled  data points .

Figure 2 Ammonium and nitrite concentrations increase with decreasing dissolved oxygen saturation.

Figure 3 Ratio of $\delta^{15}N\text{-}NO_3^-$ versus $\delta^{18}O\text{-}NO_3^-$ values corresponding to decreasing nitrate concentrations from 13 to 20 June and filled data points of figure 1b and 1c. The calculated linear regression has a slope of 0.82 with R² of 0.96.

Figure 4 Nitrite isotope values versus the remaining fraction of nitrite  during the Elbe flood corresponding to the filled data points in figure 1b and 1c. The dashed line indicates the apparent isotope effect during nitrite consumption with a slope of -10.0±0.1‰ and R² of 0.97.

Figure 5 Dual nitrate isotope values versus the remaining fraction of nitrate corresponding.  to the filled data points in figure 1b and 1c. The solid line indicates the apparent isotope effect during nitrate consumption with a slope of  $^{15}\varepsilon$ -4.0±0.1‰ with R² of 0.89 and the dashed line  is $^{18}\varepsilon$ -5.3±0.1‰ with R² of 0.92.

---

## Referee Report (RR1)

**"Isotope effects of coupled nitrification and denitrification during a river flood event", by Juliane Jacob, Tina Sanders, and Kirstin Dähnke**

The main points raised in previous reviews have been addressed. I still disagree with some of the argumentation (e.g., steady-state assumption during a flood), but overall the manuscript has improved a lot by avoiding overstatements and discussing a broader view of N cycle processes. In my opinion, the main point of this paper should not be to report isotope effects of single processes (because it is simply impossible to do so with the data in hand), but to present the isotope data (which is a truly nice set of data) and discuss the processes that led to those signatures (like it is done in the 3 scenarios). You can present numbers of apparent isotope effects, but make sure to be clear at all times that those are at best (i.e., steady-state assumption is true etc.) apparent isotope effects of net nitrate and nitrite consumption (including co-occurring production processes). Accordingly, I suggest not to state in the title that reporting isotope effects is the main point of the article.

Minor comment:

I suggest to report d18O:d15N enrichment as is usually done, and not vice versa (e.g., page 7). And accordingly plot d15N on x-axis and d18O on y-axis in fig. 3.

Check for typos, specifically regarding citations in the text.

---

## Author Response (AR2)

**Point-by-Point-Reply to „Nitrification and Nitrite Isotope Fractionation as a Case Study in a major European River" by Juliane Jacob et al. (2016)**

We thank the anonymous reviewer for the re-assessment and the positive evaluation of our manuscript.

As suggested in the first revision, we have re-focused from a certain fractionation factor towards an apparent isotope effect during nitrite and nitrate decrease, which can include production (e.g. page 9, line 3 and 27). To emphasize this further according to the reviewer's suggestions, we now stress this on page 6 line 8 – 9 and page 7 line 22 – 23. We also changed the wording to "apparent isotope effect during net nitrite/nitrate consumption" throughout the manuscript.

Furthermore, the title has been changed to "Nitrite consumption and associated isotope changes during a river flood event", because we agree that the focus of the revised version is indeed not isotope effects but unusual ammonium and nitrite concentrations and isotope variations as such.

We also followed the reviewer's suggestion to change the axis of figure 3 and the accompanying text (page 6 line 2, page 7 line 27, figure captures).

Furthermore, we thoroughly checked the manuscript and references for typos.

[revised manuscript text omitted]

**Figure 1a, b, c**

[Figure]

**Figure 2**

[Figure]

$\delta^{15}N\text{-}NO_3^- : \delta^{18}O\text{-}NO_3^- = 1.22, R^2 = 0.95$

5    **Figure 3**

[Figure]

**Figure 4**

[Figure]

**Figure 5**

**9 Figure Captions**

Figure 1a Discharge, dissolved oxygen concentration, and SPM concentration of the Elbe River water samples from 6 to 20 June 2013. Flood conditions occur with discharge >3000 $m^3 s^{-1}$.

Figure 1b Ammonium, nitrite, and nitrate concentrations in the Elbe River in the course of the flood. Calculation of the fractionation factor is based on filled data points.

Figure 1c Ammonium, nitrite, nitrate, and SPM isotope values in the course of the flood. Calculation of the fractionation factor is based on filled data points.

Figure 2 Ammonium and nitrite concentrations increase with decreasing dissolved oxygen saturation.

Figure 3 Ratio of $\delta^{15}N\text{-}NO_3^-$ versus $\delta^{18}O\text{-}NO_3^-$ values corresponding to decreasing nitrate concentrations from 13 to 20 June and filled data points of figure 1b and 1c. The calculated linear regression has a slope of 1.22 with R² of 0.95.

Figure 4 Nitrite isotope values versus the remaining fraction of nitrite during the Elbe flood corresponding to the filled data points in figure 1b and 1c. The dashed line indicates the apparent isotope effect during net nitrite consumption with a slope of -10.0±0.1‰ and R² of 0.97.

Figure 5 Dual nitrate isotope values versus the remaining fraction of nitrate corresponding to the filled data points in figure 1b and 1c. The solid line indicates the apparent isotope effect during net nitrate consumption with a slope of $^{15}\varepsilon$ -4.0±0.1‰ with R² of 0.89 and the dashed line is $^{18}\varepsilon$ -5.3±0.1‰ with R² of 0.92.

---

## Author Response (AR3)

**Point-by-Point-Reply to "Nitrite consumption and associated isotope changes during a river flood event"**

Dear Helge Niemann,

thank you very much for the revision and the positive evaluation of our manuscript.

We have changed P3, L23; P4, L20; the tense in the results chapter; P7, L33 and the order of d18O:d15N as suggested.

On page 4 line 15, we report the analytical error of our isotope measurements, which are based on triplicate standard and duplicate sample analyses.

In the sentence on page 8 line 4, we refer to the situation in soils, not in the water column or in river sediments; we do assume that the role of filtering of ammonium by clay minerals in the water column is negligible in our study. The reasons are the following: (1) the suspended matter in the water column during the flood event is relatively low and has a high content of organic matter, and, consequently, does not contain a large amount of clay minerals. This is reflected in the sediment structure. Sediments at the weir are relatively sandy with only little clay content. (2) Even if these sediments were resuspended, any present clay minerals should be loaded with exchangeable ammonium beforehand. They might then actively exchange ammonium, but this will not have a net concentration effect.

To focus on the main storyline of the manuscript, we for now refrained from further discussion of the role of clay in the water column and rephrased the sentence to "because ammonium molecules are positively charged and thus tightly bound to clay particles in soil, and elution with discharge generally does not occur".

Page 8 line 21 has been changed to "because nitrite is generally not abundant in the catchment and is immediately oxidized".

Best wishes,

Juliane Jacob